# High-order elastic multipoles as colloidal atoms

Bohdan Senyuk[1], Jure Aplinc[2], Miha Ravnik [2,3] & Ivan I. Smalyukh [1,4,5]

Achieving and exceeding diversity of colloidal analogs of chemical elements and molecules as building blocks of matter has been the central goal and challenge of colloidal science ever since Einstein introduced the colloidal atom paradigm. Recent advances in colloids assembly have been achieved by exploiting the machinery of DNA hybridization but robust physical means of defining colloidal elements remain limited. Here we introduce physical design principles allowing us to define high-order elastic multipoles emerging when colloids with controlled shapes and surface alignment are introduced into a nematic host fluid. Combination of experiments and numerical modeling of equilibrium field configurations using a spherical harmonic expansion allow us to probe elastic multipole moments, bringing analogies with electromagnetism and a structure of atomic orbitals. We show that, at least in view of the symmetry of the "director wiggle wave functions," diversity of elastic colloidal atoms can far exceed that of known chemical elements.

[1] Department of Physics and Soft Materials Research Center, University of Colorado, Boulder, CO 80309, USA. [2] Faculty of Mathematics and Physics, University of Ljubljana, Jadranska 19, 1000 Ljubljana, Slovenia. [3] J. Stefan Institute, Jamova 39, 1000 Ljubljana, Slovenia. [4] Department of Electrical, Computer and Energy Engineering, University of Colorado, Boulder, CO 80309, USA. [5] Renewable and Sustainable Energy Institute, University of Colorado, Boulder, CO 80309, USA. Correspondence and requests for materials should be addressed to I.I.S. (email: ivan.smalyukh@colorado.edu)

Colloids are ubiquitous in everyday life and can be found everywhere from industrial, highly technological materials to commonly used health care and nutrition products[1–3]. Macroscopic structure and physical properties of self-assembled colloidal systems can be tuned by changing the interaction between their building blocks. Thus, designing new colloidal particles as atom-like building blocks can enable self-assembly of new complex artificial materials with desired properties, including the ones not encountered in nature. This "bottom-up" approach caused an explosion in the development of various kinds of colloidal particles[1–5]. Geometry of colloidal particles is often used to define directional interactions between them, similar as the valence of atoms determines bonds that atoms can form within molecules and crystals[6]. Directional bonding of colloidal particles has been pursued in many systems of colloidal "atoms" with varying shape and geometry[6], patchy features[7–9], and via various chemical functionalizations and DNA-hybridization[7,10–14] of nanoparticles and microparticles. Elastic multipoles induced by colloidal particles dispersed in liquid crystals (LCs) provide yet another promising approach of defining controlled directional interactions of nematic colloidal "atoms"[15–31], with design principles often building on the analogy of these multipoles with their electrostatic counterparts. When designing self-assembly of colloidal superstructures, conditions of certain multipolar charge distributions, as well as the nature of interactions between them, can provide insights into how nematic colloids can be controlled[15]. For example, similar to electrostatic charge distributions, odd moments of elastic multipoles are expected to vanish when the LC director field $\mathbf{n}(\mathbf{r})$, which describes spatial patterns of orientation of rod-like constituent molecules, is symmetric about the particle center and a plane orthogonal to the far-field director $\mathbf{n}_0$, as in the cases of elastic quadrupoles[17–19] and hexadecapoles[31]. On the other hand, both odd and even moments should be present for particles with asymmetric $\mathbf{n}(\mathbf{r})$, such as elastic dipoles[16,18]. The design of elastic colloidal multipoles can potentially not only take advantage of such symmetry considerations, but also build on the versatile means of controlling $\mathbf{n}(\mathbf{r})$ by surfaces of colloidal inclusions with elaborate geometry and topology[15]. However, general physical principles and feasibility of "on-demand" achieving a diverse variety of elastic colloidal multipoles remain unknown.

Colloidal particles locally distort the uniform alignment of $\mathbf{n}(\mathbf{r})$ in a nematic LC host medium, prompting elasticity-mediated interactions between them, which tend to minimize the system's free energy[16]. Surface boundary conditions on colloidal particles play an important role, often introducing bulk and surface line and point defects[15–31]. This gives rise to interactions between colloidal particles, which tend to arrange such that energetically costly distortions can be shared, and essentially resemble interactions of electrostatic multipoles[16,18–21]. However, mainly only colloidal elastic dipoles[22,23] and quadrupoles[17,22,24,25] were studied, whereas the higher order multipoles were rarely considered, although recently the conic degenerate anchoring boundary conditions[30] enabled observation of hexadecapolar (16-pole) LC colloids[31]. Beyond the electrostatic analogy, elastic multipoles also share the mathematical description in terms of spherical harmonics with chemical elements, whereby the elastic monopoles, dipoles, quadrupoles, and octupoles have atomic analogs with the structure of filled *s*-, *p*-, *d*-, and *f*-orbitals[15,31]. Since none of the known chemical elements have filled orbitals higher than *f*, colloidal "atoms" in the form of elastic hexadecapoles and higher order multipoles have the potential to go beyond and, thus, could provide completely new insights and means of realization of new breeds of composite materials. Moreover, such artificial atoms and molecules could be used not only to define symmetry and structure of mesoscopic composite material systems, but also their physical properties. Indeed, the analysis of electric and magnetic multipoles[32] is also performed in optical metamaterials, as a central way to characterize the interaction of the electromagnetic fields with the material, which today underpins some of the most important technologies, ranging from telecommunications to data storage and light-assisted manufacturing[33,34]. Within this approach, electromagnetic media are described as a set of point-like multipole sources, consisting of electric, magnetic, and toroidal multipoles[33,34]. Such metamaterials are typically nanofabricated, which limits their utility and calls for the development of means to self-assemble them from colloidal meta-atoms.

In this work, we uncover the physical mechanisms that may allow for an on-demand control of the leading-order elastic multipoles in LC colloids. We systematically demonstrate how shape and boundary conditions on colloidal particle surfaces determine the structure of elastic distortions, which allows us to identify the "design rules" for obtaining desired elastic multipoles and, thus, also the ensuing elasticity-mediated interactions. This may allow for developing preprogrammed composite metamaterials that self-assemble to yield the desired mesoscopic structure and physical properties.

## Results

**Elastic multipoles in nematic LCs.** Multipole expansion is an approach that represents a distinct complex spatially varying field, electric, magnetic, gravitational, material, as a series of elementary contributions—the multipoles[32,33,35]. Multipoles represent the magnitude and spatial profile of basic sources of the fields, typically in some small region, to give full fields in more distant regions. Multipole moments usually consist of inverse powers of the distance from the sources and angular dependent terms with the key assumption that only some, usually selected lowest order, multipoles are sufficient to adequately describe the full variability of considered fields. The existence or nonexistence of some multipoles not only has profound importance for the properties of the fields, but can even determine fundamental laws. One prime example is the nonexistence of magnetic monopoles, which fundamentally determines the structure of Maxwell's equations of electromagnetism, one of the fundamental laws of Nature[36]. Another prime example are atomic orbitals in atoms, where quantum numbers of orbitals reflect the corresponding underlying multipole-type nature of atoms[37].

We explore elastic multipoles in the material orientational field of nematic complex fluids, with a central distinction that these multipoles can be directly measured and determined with optical and material science techniques and, moreover, that usually noncommon leading-order multipoles such as 16-pole, 32-pole, and even 64-pole can be realized (Fig. 1). The ordering field of nematic complex fields, into which multipoles will be imprinted by colloidal particles of designed surface-imposed ordering, is determined by an effective total free energy, which in full Landau-de Gennes form consists of effective nematic bulk elastic and ordering and surface anchoring terms[38] (Methods). The leading contribution that can transfer interaction, e.g., between multiple colloidal particles, colloidal atoms, in analogous way as electromagnetism in atomic orbitals, is the nematic elasticity which in the elementary one-elastic-constant form can be written as

$$f_E = \frac{1}{2}K \sum_{\mu=x,y} \left(\nabla n_\mu\right)^2, \tag{1}$$

where $f_E$ is elastic free energy density, $K$ is the single-average Frank elastic constant and $n_\mu$ ($\mu = x, y$) are director components perpendicular to the far field direction (*z*-axis). This formulation of the free energy density relies on the crucial assumption of roughly uniform director field $\mathbf{n}(\mathbf{r}) \approx (n_x, n_y, 1)$, with small

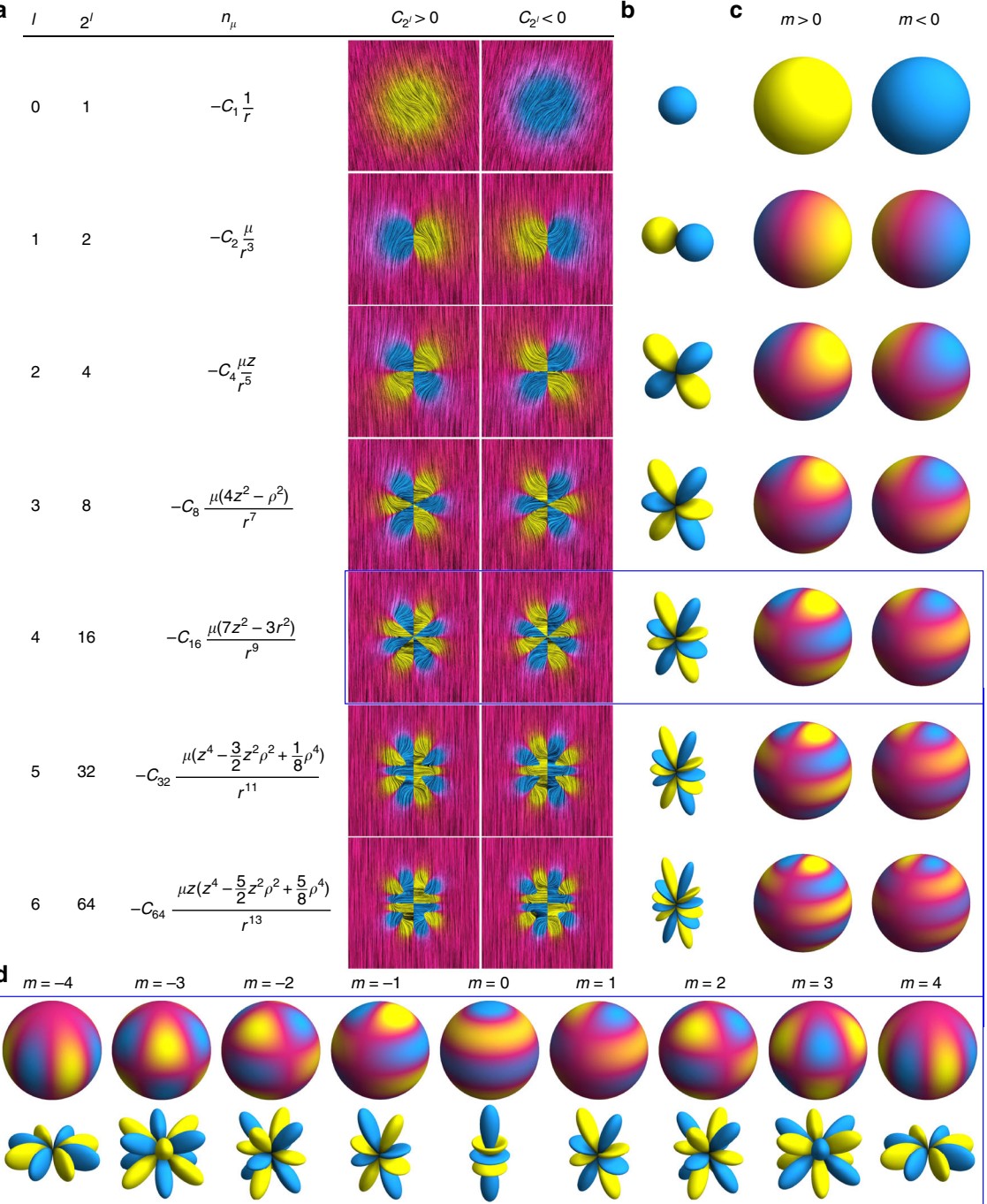

**Fig. 1** Correspondence between elastic multipoles, electrostatic charge distributions, and atomic orbitals. **a** Analytic elastic multipoles. The $xz$ cross-section of the normalized director field $(n_x, n_y, 1)$ of individual multipoles in Cartesian coordinates. Note that $r^2 = x^2 + y^2 + z^2$ and $\rho^2 = x^2 + y^2$. **b** Diagrams for $s$-, $p$-, $d$-, $f$-, $g$-, $i$- and $k$-atomic orbitals calculated using the angular wavefunction. **c** Elastic multipoles around spherical particles with a tilt of director at their surface. **d** Elastic multipoles and atomic orbitals for hexadecapoles with different $m$

$n_x$, $n_y \ll 1$, which becomes justified at large distances from the localized source of nematic distortions, such as colloidal particles. The elementary solutions of the nematic field around localized sources, elastic multipoles, can be introduced by minimizing the free energy density (Eq. (1)) with the Euler–Lagrange formalism giving Laplace equations

$$\nabla^2 n_\mu = 0, \qquad (2)$$

which in full 3D are separable and can be analytically solved as a

series, i.e., as a summation over the elastic multipoles

$$n_\mu(r, \theta, \phi) = \sum_{l=0}^{\infty} \sum_{m=-l}^{+l} q_{lm}^\mu \frac{R_{\text{eff}}^{l+1}}{r^{l+1}} Y_l^m(\theta, \phi), \qquad (3)$$

where $\theta$ is polar and $\phi$ azimuthal angle, $Y_l^m(\theta, \varphi)$ are spherical harmonics, $q_{lm}^\mu$ are dimension-free elastic spherical multipole coefficients, $l$ determines the order of a multipole as $2^l$th pole, $-l \leq m \leq l$, and $R_{\text{eff}}$ is the characteristic scale of the multipole (given in our case by the effective size of the particle). Using orthogonality of spherical harmonics, multipole moments $q_{lm}^\mu$ can

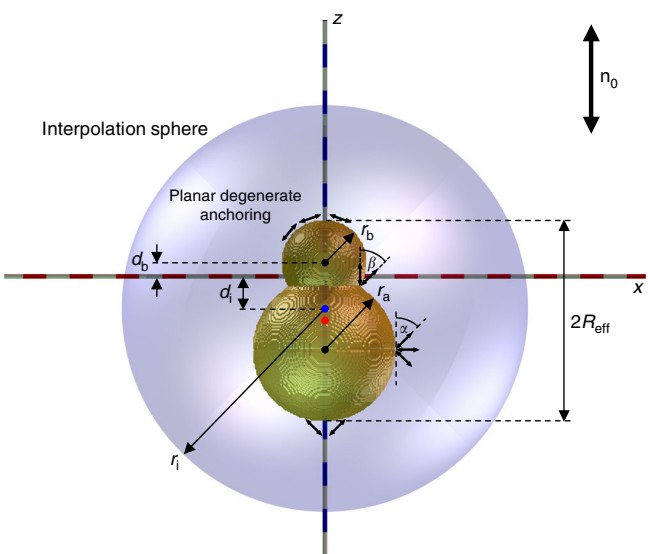

**Fig. 2** Model of a colloidal particle consisting of two interpenetrated spheres used in calculations. The lower sphere has a constant radius $r_a$ and is positioned at a distance $-r_a$ from the coordinate origin (note axes $x$ and $z$ are indicated by dashed red and blue lines, respectively). The upper sphere's radius $r_b$ and distance $d_b$ of the upper sphere center from the origin are particle geometrical parameters varied in the process of analysis. The anchoring on the upper sphere is planar degenerate, while that on the lower sphere is conic degenerate with a tilt angle $\alpha$. At the neck, where two spheres with distinct anchoring meet, the angle between two anchoring directions is represented with angle $\beta$. Blue sphere indicates the interpolation sphere of radius $r_i$ and blue dot depicts its center displaced by $d_i$ from the origin, both $r_i$ and $d_i$ are varied in the analysis to determine the center and magnitude of the elastic multipoles. Red dot represents geometrical center of the composite colloidal particle whereas half of the composite particle length along $z$-axis represents the effective particle radius $R_{eff}$

be determined with the following integral

$$q_{lm}^\mu = \int_0^{2\pi} \int_0^\pi n_\mu(r,\theta,\varphi) \frac{r^{l+1}}{R_{eff}^{l+1}} Y_l^{m*}(\theta,\varphi) d\theta d\varphi, \qquad (4)$$

which is particularly useful for calculating multipoles numerically. Clearly, this integration must be performed at the radius that is large enough to satisfy assumptions of $n_x$, $n_y \ll 1$ and $n_z \approx 1$. In practice the director field $n_\mu$, obtained from experiments or modeling, is defined only in discrete points $(\theta_i, \phi_j)$; therefore, the calculation of multipole coefficients reduces to the discrete Fourier and Legendre transform on a selected spherical grid (Methods). Figure 1 shows examples of elastic multipoles and relates the symmetry of associated director distortions to the analogous descriptions of charge distributions in electrostatics and electron wave functions in the description of electron shells of chemical elements.

**Elastic multipoles at composite colloids of dissimilar spheres.** The aim of this work is to systematically investigate how high-order elastic multipoles can be induced by colloidal particles with varying shape and boundary conditions. Dimers of spheres (Fig. 2) are interesting sources of elastic distortions because they can be mass-synthesized using wet chemistry approaches[39–41] (making them relevant for composite material fabrication) while also allowing for more complex director distortions than what can be induced by individual colloidal spheres. We first study

elastic multipoles formed around colloidal particles comprised of two dissimilar spheres (Figs. 3, 4) having different sizes, composition, and anchoring. In the first case, gourd-shaped dimer colloidal particles consist of two lobes of different diameter and with different surface anchoring boundary conditions for **n(r)** (Fig. 3a–f). The dissimilar anchoring is defined through the particle synthesis, in which cross-linked polystyrene spherical seeds (a smaller lobe) were swollen with styrene and the elastic contraction of the cross-linked polystyrene expels styrene out of the swollen seeds to give rise to the second (larger) lobe[39–41]. As a result, the smaller particle's lobe has tangential anchoring and the larger lobe has a conic anchoring[30,31]. Figure 3a–c shows POM textures of such particles in LC, from which the director structure around the particle (Fig. 3d) was experimentally deduced. Gourd-shaped dimers align with their cylindrical symmetry axis parallel to **n₀**. These nematic colloids induce two surface point defects (called "boojums"), one at the south pole of a large lobe and another at the north pole of a smaller lobe, as well as a surface defect loop at the equator of the larger lobe (Fig. 3d, e). In addition, a singular defect loop is visible spanning around the neck of a dimer particle, where two lobes come to the contact, which is due to the mismatch in the alignment of molecules at both lobes in the point of contact. Interestingly, the presence of this defect loop can be avoided by adjusting a distance $d_b$ (Figs. 2, 5) so that the particle geometry is fully compatible with boundary conditions at the two lobes, as we will show below using numerical calculations. The induced configuration of **n(r)** is complex and the number of reversals of the director tilt with respect to **n₀** at the surface of the particle is the same as in an elastic 64-pole (compare Figs. 3d, e and 1 for $l = 6$). However, the structure also lacks symmetry with respect to the plane orthogonal to **n₀**, which is not the case for the pure elastic 64-pole (see the analytical ansatz in Fig. 1). While the strengths of different elastic multipoles for such particles can be assessed numerically (see below), also the elastic interactions between such gourd-shaped dimers provide important insights (Fig. 3h–j). The highly anisotropic elastic interactions depend on the multipole magnitude and orientation (with respect to **n₀**) of the separation vector connecting centers of two interacting gourd-like dimers. There are many narrow zones of attractive interactions separated by zones of repulsive interactions (Fig. 3g). For a colloidal particle with a 64-pole strongly dominating (or pure), the angular dependence would consist of 24 such zones, 12 attractive, and 12 repulsive (Fig. 3g). However, the experimental angular diagram (Fig. 3h) is even more complicated because of the presence of other multipole moments, in addition to the 64-pole. Depending on the orientation of the separation vector with respect to **n₀**, gourd particles self-assemble into various pair arrangements, examples of which are shown in Fig. 3k–o. This rich behavior illustrates that the tunable multipolar nature of elastic high-order multipoles can be used for predefining colloidal self-assembly structures. Furthermore, these findings are consistent with the particle's lack of symmetry plane orthogonal to **n₀** and numerical results presented for colloidal particles with tunable shape that we discuss below. The elastic pair interaction potential between colloidal particles extracted from the experimental distance vs. time dependencies is several hundreds of $k_B T$, with the interaction force approaching ~1 pN near their full contact. Fitting the experimental interaction potentials (Fig. 3i, j) with an expression for the colloidal pair-interaction energy within the multipolar approach[31]

$$U_{int} = 4\pi K \sum_{l,l'} b_l b_{l'} (-1)^{l'} (l+l')! \frac{R_{eff}^{l+l'+2}}{r^{l+l'+1}} P_{l+l'}(\cos\theta),$$

where $P_{l+l'}(\cos\theta)$ are the Legendre polynomials, yields the coefficients corresponding to the strength of elastic multipole moments (Fig. 3). We find that the hexadecapolar moment $b_4$ is

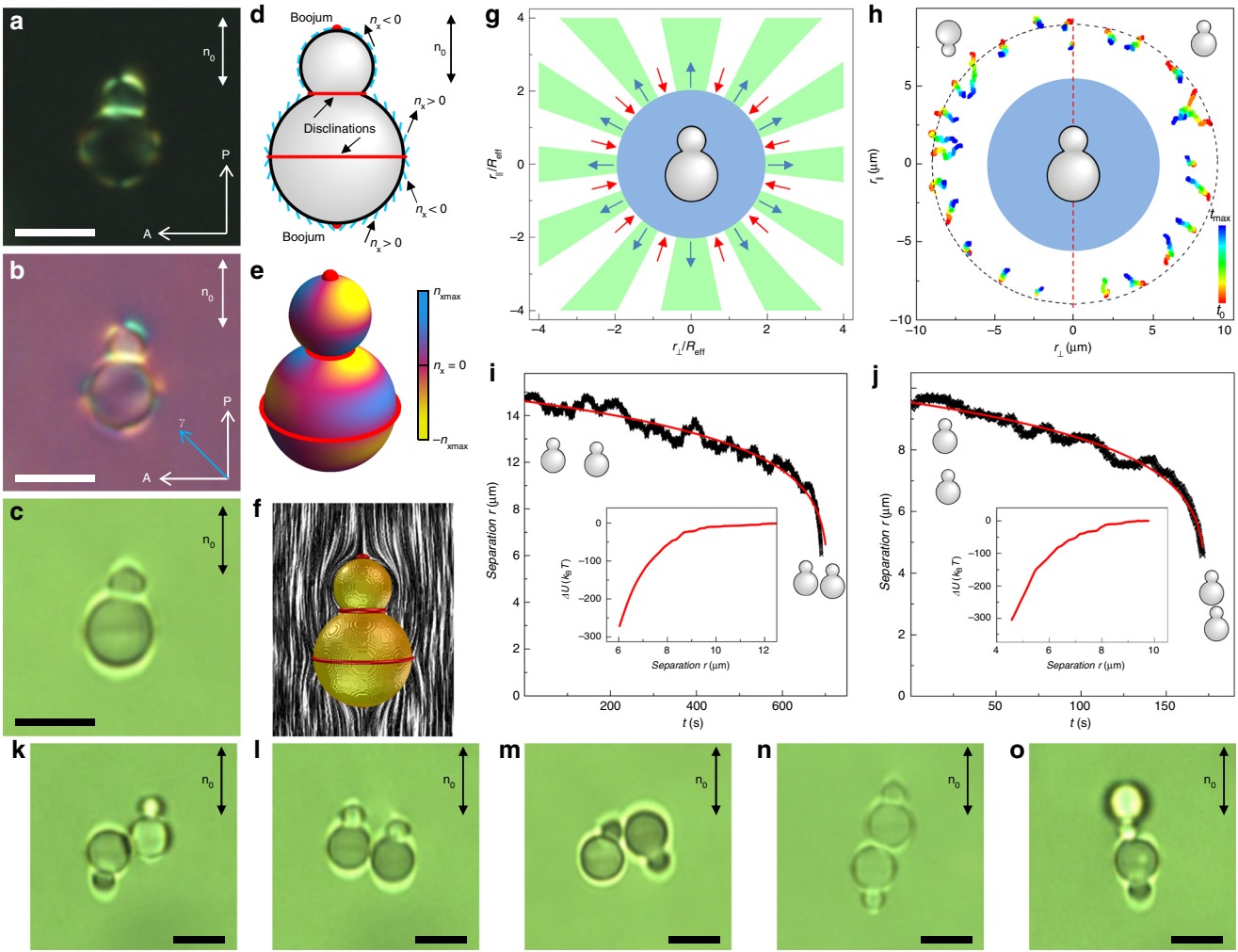

**Fig. 3** Elastic multipole induced by a gourd-shaped particle. **a–c** Textures of a sample with a gourd-like particle in a nematic cell from optical polarizing, **a** without and **b** with a retardation plate, and **c** bright field microscopy. **d**, **e** Schematic diagram (blue lines) of $\mathbf{n}(\mathbf{r})$ at the surface and corresponding color-coded diagram of $n_x$. **f** Calculated $\mathbf{n}(\mathbf{r})$. **g**, **h** Map of elastic interactions between gourd particles with respect to $\mathbf{n}_0$ calculated for elastic 64-poles with (**g**) only 64-pole non-zero coefficient and (**h**) experimentally measured. Dashed red line in (**h**) separates maps of interactions for parallel and antiparallel particles. **i**, **j** Separation vs. time dependence for gourd particles elastically interacting along the direction approximately (**i**) perpendicular and (**j**) parallel to $\mathbf{n}_0$. Insets show the corresponding dependence of the interaction potential from the distance between particles with a 64-pole coefficient $b_6 \approx -3 \times 10^{-5}$ extracted from fitting. **k–o** Textures of self-assembled pairs of the gourd-shaped particles imaged by bright-field microscopy. Scale bar: 5 μm

still strongly pronounced, which is due to the larger lobe with conic anchoring (Fig. 6). Although different approaches can be used for fitting (Fig. 3), it is clear that the role of 64-pole ($b_6$) becomes significant at small inter-particle distances $r$. The interaction potential of gourd-shaped particles (Fig. 3i, j) cannot be fit using expressions with only quadrupolar ($b_2$) or hexadecapolar ($b_4$) nonzero coefficients. The angular dependence of elastic interactions of two gourd-shaped particles, as expressed by different relative magnitudes of the multipole coefficients, is very sensitive to the geometrical parameters of a dimer, like the relative dimensions and overlap of the dimer lobes, which is calling for the need of establishing design principles for expressing the desired leading-order multipoles, as we do numerically below.

Another model colloidal object in our study is a dimer particle consisting of two spheres with dissimilar size and anchoring boundary conditions, which is made from superparamagnetic beads (SPMB) and glass spheres (Fig. 4). SPMBs are somewhat smaller as compared to glass particles (Fig. 4). Using the laser tweezer, two different particles were placed close to each other and LC around them was locally melted to the isotropic state by converting the infrared laser irradiation to heat through

absorption, irreversibly forming a colloidal dimer bound by van der Waals forces. The LC was then quenched back to the nematic state. The resulting dimer aligns with the symmetry axis along $\mathbf{n}_0$ (Fig. 4). In bright-field microscopy textures, epoxy-based SPMBs look brownish due to the light absorption by magnetic nanoparticles embedded within them. SPMBs impose the planar alignment on the LC director. On the contrary, the treated glass particles impose perpendicular alignment of LC molecules at their surfaces. The resulting director structure contains a surface point defect on the pole of a SPMB sphere and a bulk disclination loop, called "Saturn ring", at the equator of a glass sphere (Fig. 4e–g). Analysis of experimental and numerical $\mathbf{n}(\mathbf{r})$, including the corresponding color-coded diagram of the $n_x$ sign alternation at the surface of the dimer, reveals the structural similarity with an elastic octupole with $l = 3$ (compare Figs. 4e, 1). Other structures of $\mathbf{n}(\mathbf{r})$ around such dimers were also observed (Fig. 4d, h), within which the accompanying disclination loop shifted from the equator of the glass particle and resided near the SPMB sphere, just above the contact point of two particles, effectively generating a dipolar director structure. This diversity of multipolar structures[7] that can be induced by colloidal dimers calls for

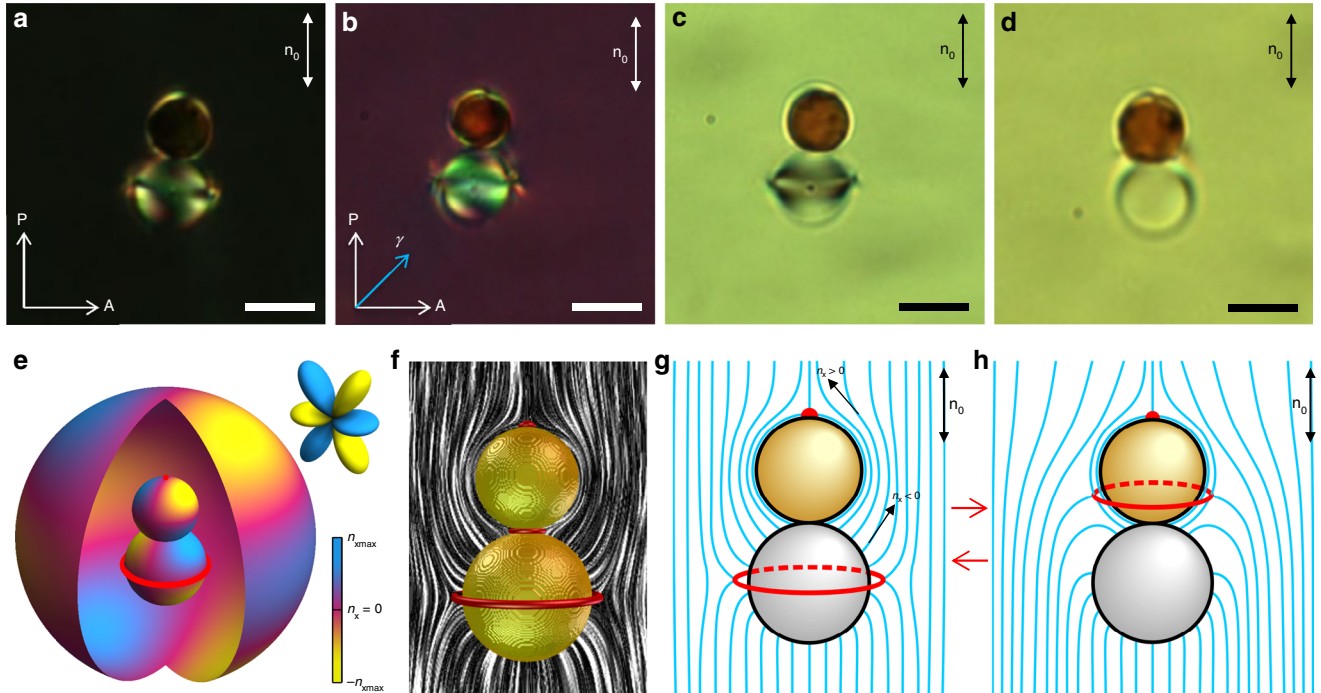

**Fig. 4** Elastic multipoles induced by a pair of dissimilar particles. **a–d** Textures of a sample with particles in a nematic cell obtained by optical polarizing, (**b**) with and (**a**) without a retardation plate, and (**c**, **d**) bright field microscopy. Top particle has planar and bottom particle has homeotropic anchoring. **e–g** Schematic diagram (blue lines) of $\mathbf{n}(\mathbf{r})$ (**g**) also calculated in (**f**) and (**e**) corresponding color-coded diagram of the $n_x$ directly at the surface of the particle shown in (**a–c**) and at the surface of the interpolation sphere. Inset shows a corresponding atomic orbital. **h** Schematic diagram of $\mathbf{n}(\mathbf{r})$ around a pair of particles shown in (**d**). Scale bar: 5 μm

identifying parameters that can lead to on-demand formation of various multipoles.

Figure 5 shows numerically calculated structures of $\mathbf{n}(\mathbf{r})$ induced by colloidal particles composed of two dissimilar spherical colloidal objects. The radius of the upper sphere $r_b$ is varied from 0 to the size of the lower sphere $r_a$ in steps of $r_a/5$. The position of the upper sphere is also varied and is gradually immersed into the lower sphere in steps of $r_a/5$, where, in the limiting regimes, the upper sphere is completely contained (left side of the pyramids) or barely touches the lower one (right side of the pyramids). Conic anchoring on the lower sphere introduces additional parameter $\alpha$ (Fig. 2) that corresponds to the tilt angle of the conic anchoring. In the limiting case when the conic anchoring angle $\alpha$ reaches 90° (Fig. 5a), the director is locally oriented perpendicular to the surface of a lower spherical lobe. This homeotropic anchoring induces a disclination loop defect "Saturn ring" on the lower sphere. The tangential anchoring on the upper sphere induces a boojum at its north pole. Another loop of reduced degree of order can emerge in the neck, where the two spheres with distinct anchoring meet. It is induced by a mismatch between the two easy-axis directions imposed at the contact of the two surfaces by two different boundary conditions, measured with angle $\beta$ (Fig. 2). The mismatch angle $\beta$ characterizes the effective surface-imposed frustration in the defect region within the neck and depends on both the conic anchoring angle $\alpha$ as well as on the angle at which the two spheres intersect. The neck region with the reduced degree of order vanishes when geometry of the composite colloid (size and displacement of the upper sphere) assures intersection of the two spheres at 90° setting mismatch angle $\beta = 0$ (third diagonal from the right side of the pyramid in Fig. 5a). As also observed in experiments (Fig. 4d, h ), the Saturn ring can change its vertical position and slide from equator of the lower colloid toward the neck, or even on the neck joining with the regular neck defect. This rearrangement occurs in the lower left

region of the parameter–space pyramid (Fig. 5a), where $\mathbf{n}(\mathbf{r})$ is found to be dipolar-like.

Figure 5b–d shows the composite colloids and their corresponding field structures when conic anchoring boundary conditions respectively at angles $\alpha = 60°$, $\alpha = 40°$, and $\alpha = 20°$ are applied on the lower sphere. For these conic angles, the distinct nature of conic anchoring (Fig. 2) induces a ring defect at the equator of the lower sphere. The tangential anchoring on the upper sphere establishes a boojum defect at the surface of the colloidal particle. At the neck, where the two spheres intersect and the two regions with dissimilar anchoring meet, again the singular region of reduced order emerges. Its actual profile depends on the mismatch angle $\beta$, which is now conditioned by the conic anchoring angle $\alpha$ and composite colloid parameters $r_b$ and $d_b$. The neck region of reduced order completely vanishes when the two spheres intersect at an angle equal to $\alpha$. Indeed, one can see that the neck region of reduced order effectively disappears for the composite colloids at third diagonal from the left side of the pyramid in cases $\alpha = 60°$ (Fig. 5b), $\alpha = 40°$ (Fig. 5c) and for second diagonal from the left in case of $\alpha = 20°$ (Fig. 5d). When $\alpha$ is small (Fig. 5d), the anchoring on the lower colloid is nearly tangential. Consequently, the effective Saturn ring distortions become less pronounced, whereas distortions of the neck region of reduced order becomes stronger at large separations $d_b$ (right edge of the pyramid on Fig. 5d). We have analyzed elastic multipole moments of experimental (Figs. 3, 4) and simulated (Fig. 5) structures of composite particles using decomposition on spherical multipoles with the SHTns numerical library (Methods), as detailed below.

Figure 6 shows spherical multipole coefficients of all simulated composite colloids at various angles $\alpha$. The dipolar coefficient $q_{11}^x$ is generally weak and positive for low $\alpha$; however, a prominent peak is observed at $\alpha = 90°$ for the structures close to $r_b/r_a = 4/5$, $d_b/r_a = -2/5$ (see Fig. 5a). Consistent with experiments, it is

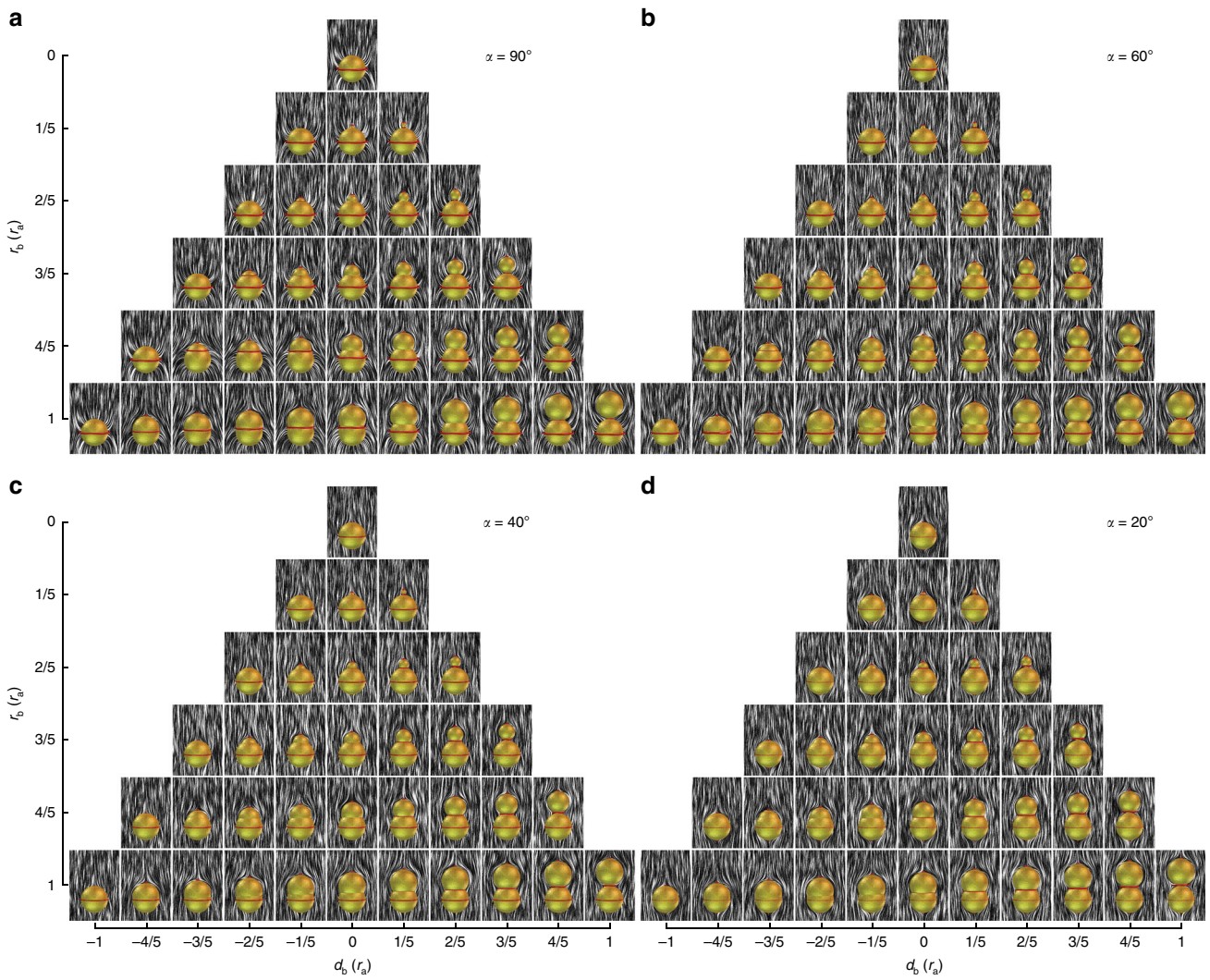

**Fig. 5** Gourd-like colloids and corresponding director field structures. Upper sphere has planar degenerate anchoring and lower sphere has tilted anchoring with various angles $\alpha$. Pyramid-like diagrams show composite colloidal particles (yellow isosurfaces) with variable radius $r_b$ and position $d_b$ of the upper sphere and conic anchoring at angle: **a** $\alpha = 90°$, **b** $\alpha = 60°$, **c** $\alpha = 40°$, and **d** $\alpha = 20°$ on the lower sphere. Defect regions are indicated with red isosurfaces ($S = 0.44$), whereas the director field is illustrated with black and white streaks. **a** Defects generally emerge at the lower sphere equator, upper sphere north pole and possibly at the neck where both spheres intersect. The Saturn ring surrounding the lower sphere can move upwards or even to the neck, joining with the neck defect (lower left side of the pyramid). **b**, **c** Defects generated by conic anchoring emerge precisely at the equator of the lower sphere, whereas planar degenerate anchoring induces boojum defect at the north pole of the upper sphere. The mismatch between the two anchoring directions forms the surface neck defect. The profile of the neck defect is varied by changing the upper sphere geometry and position; it completely vanishes for composite colloids on the third diagonal from the left side of the pyramid. **d** The neck defect completely vanishes for composite colloids on the first diagonal from the left side of the pyramid

energetically favorable that the Saturn ring defect moves from the equator of the lower particle to the neck, effectively joining with the neck defect. Defect in the neck only weakly distorts the surrounding director field (Fig. 5a); therefore, the defect playing a key role is the boojum at the north pole of the upper particle, resulting in a dipolar director deformation. The dipolar multipole is still present in this region, even at lower $\alpha$. The quadrupolar coefficient $q_{21}^x$ is strongest and negative for homeotropic bottom particles, where the Saturn ring is the sole defect, and becomes weaker upon decreasing $\alpha$. At $\alpha = 40°$ negative contribution vanishes and strong positive quadrupole moment emerges for spherical colloids with nearly tangential anchoring at $\alpha = 20°$, where two boojums emerge at both poles. The transition between a positive- and negative-quadrupole moments can be understood by comparison with Fig. 1. An octupole moment $q_{31}^x$ is always positive and strongest for large upper spheres and conic

anchoring angles near $\alpha \sim 90°$. This composite colloid has the Saturn ring on the lower colloids equator, and a strong boojum on the upper particle's north pole (Figs. 4, 5a), whereas the role of neck defect in defining multipoles often can be negligible. The hexadecapole moment $q_{41}^x$ is positive and has similar strength and pattern at all angles $\alpha$. The hexadecapolar contribution of the homeotropic spherical particle emerges because the lower particle is off-centered within the simulation cell, which affects the high multipoles. The hexadecapolar coefficient is present also at angles other than $\alpha = 90°$. The best candidates for "pure" hexadecapolar elastic colloids are spheres with conic anchoring, which preserves the Saturn ring, but also induce boojums at the poles, which is in good agreement with the recent experiments[31]. The high-order 32-polar coefficient $q_{51}^x$ is generally weak. It is the strongest for composites with larger upper sphere and high-anchoring angle, which produces the Saturn ring on the lower particle and boojum

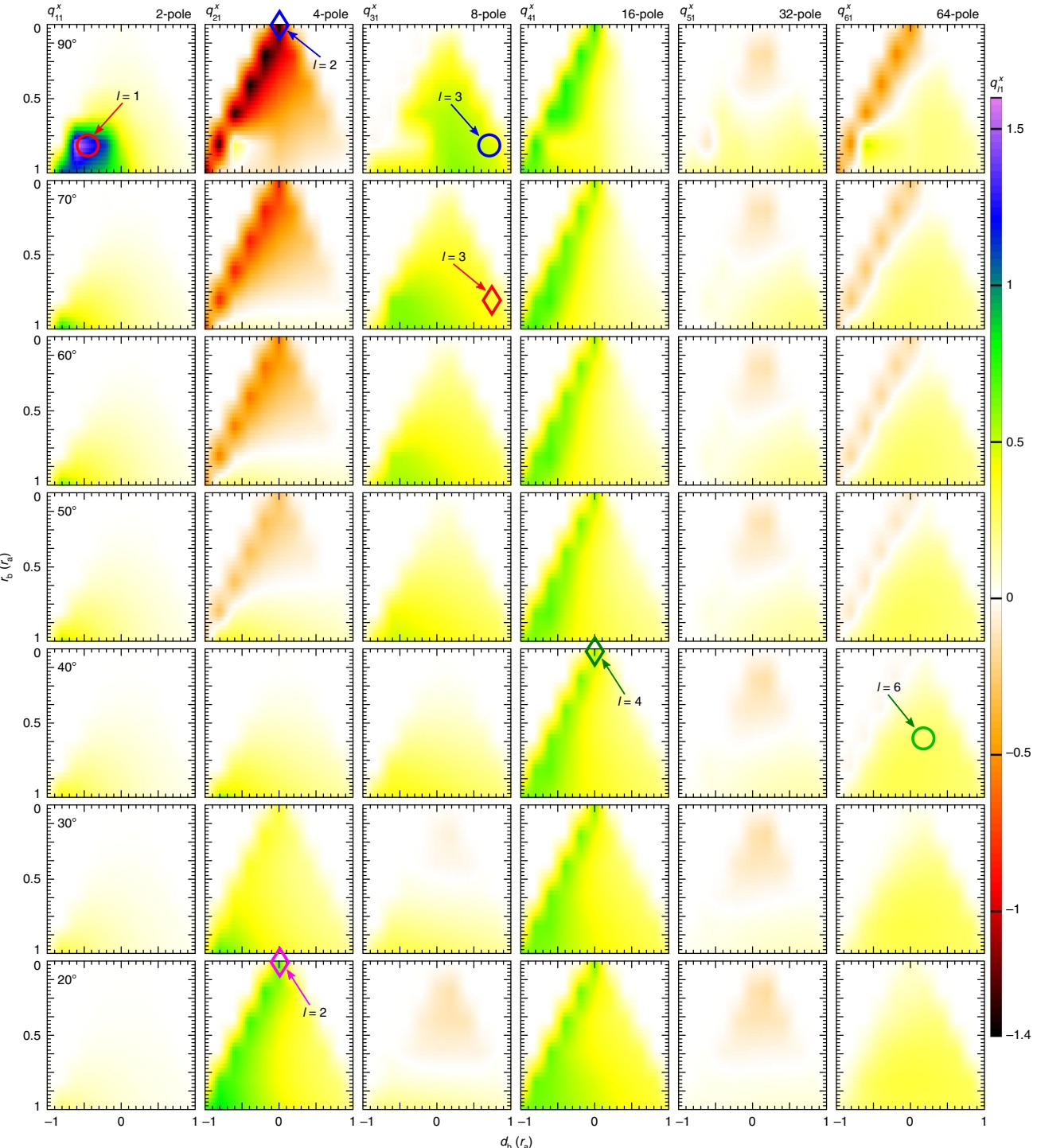

**Fig. 6** Spherical multipole moments of composite gourd-like colloids. Values of $q_{l1}^x$ ($l = 1$–6) were calculated at the particles geometrical center for all simulated composite gourd-like colloids with various conic anchoring angles at the lower lobe. Strong dipole emerges for particles near $r_b = 4/5$, $d_b = -2/5$, quadrupole is most pronounced for a spherical particle with homeotropic or nearly tangential boundary condition. Octupole is larger for composites with large upper spheres and conic anchoring, whereas the strong hexadecapole emerges for spherical particles with conic anchoring on the surfaces. 32- and 64-pole are generally weak compared to the others because the neck defect region only slightly perturb **n(r)**. Note that the multipole moments are calculated only in discrete points and interpolated for better visibility. Circles point to multipole moments of configurations with parameters close to experimentally realized: red circle corresponds to a dipole in Fig. 4d, h , blue circle corresponds to the colloid with an octupolar configuration in Fig. 4a and green circle corresponds to a 64-pole in Fig. 3. Rhombs point to the diagram regions with dominant or "pure" multipole moments: blue and magenta rhombs show respectively quadrupoles with a Saturn ring and boojums; red and green rhombs point, respectively, to dominant octupole and hexadecapole

at the north pole. The 64-polar coefficient $q_{61}^x$ appears to be most prominent for spherical particles with homeotropic boundary conditions. As in case of the hexadecapole, it is related to the strong dipole and quadrupole. Besides the simple homeotropic sphere, the 64-polar contribution is maximized for composites with large upper sphere and moderate anchoring angle $\alpha$, which provides a boojum at the upper particle, a neck defect, a Saturn ring at the equator of the lower particle and boojum at the lower particle (Fig. 3).

By tuning the geometry and anchoring on the composite particle with two dissimilar spheres one can maximize certain multipole moments and suppress the others. For example, a homeotropic particle with dimensions near $r_b/r_a = 4/5$, $d_b/r_a = -2/5$ acts as a strong dipole. Expectedly, colloidal spheres with homeotropic or nearly tangential anchoring exhibit a strong quadrupolar moment. An octupole moment is maximized for composite particles, which consists of a large upper sphere with tangential anchoring touching the lower sphere with homeotropic boundary conditions (Fig. 4). A hexadecapole moment is the strongest for single spherical particles with conic anchoring at $\alpha = 40°$.

The comprehensive multipolar analysis presented in Fig. 6 shows that different leading-order multipoles can be created simply by engineering distributions of director field, closely mimicking the corresponding analytical ansatzes (Fig. 1). Clearly, to have a desired leading order $2^l$-pole with $m = 1$ and structure axially symmetric with respect to $\mathbf{n}_0$, the director tilt (and the sign of $n_x$) must change $2^l$-times when one circumnavigates around the colloidal object once (Fig. 1), with additionally the structure being symmetric with respect to a plane orthogonal to $\mathbf{n}_0$ for all even-order multipoles. With color presentations of director tilt (Fig. 1), it suffices to count the yellow-blue changes of colors to assure that this is the case. Our findings also reveal how higher-order multipole can be expressed by introducing additional defects, either boojums or disclination rings. For the case of dissimilar dimer particles, our analysis shows that a structure with one boojum tends to exhibit a leading-order dipole moment ($l = 1$, $2^l = 2$), whereas presence of a Saturn ring tends to yield a quadrupole ($l = 2$, $2^l = 4$) (see Fig. 1). The octupole ($2^l = 8$) is produced by combining the boojum ($2^l = 2$) and the Saturn ring ($2^l = 4$) (Fig. 4): $2 \times 4 = 8$, while hexadecapole ($2^l = 16$) can be thought of as a superposition of two quadrupoles ($2^l = 4$): $4 \times 4 = 16$ or comprising a quadrupole ($2^l = 4$) and two boojums ($2^l = 2$): $2 \times 4 \times 2 = 16$. These examples yield one recipe to generate an arbitrary high-order multipole $M$ ($M$-pole) by the dissimilar dimer particles

$$M = 2^i \times 4^j, \tag{5}$$

where $i$ is the number of boojums and $j$ is the number of Saturn ring defects. Here, we consider defects solely as building blocks of director structures at which the vectorized $x$-component of the director changes its sign (Fig. 1). Also, we disregard the fact that, in general, beside boojums and Saturn rings, point defects (e.g., such as hyperbolic hedgehog) can form in nematic colloids, for which one would expect that they break symmetry in multipoles in analogous way as boojums. Finally, the detailed numerical exploration of strengths of multipole moments (Fig. 6) and the experimental example shown in Fig. 3 also show that for desired multipoles to dominate also geometric parameters and boundary conditions need to be optimized to reduce the other competing multipole moments. The symbols like circles and diamonds in Fig. 6 mark regions in the parameter space of colloidal dimers where such pure or dominant leading-order multipoles can be achieved.

**Elastic multipoles at composite colloids of similar spheres.** Dimer particles studied above are examples from a large family of colloidal objects that can be used to induce desired elastic multipoles. As another experimentally accessible example, we study composite colloidal inclusions consisting of spherical constituents with similar size and boundary conditions. We use epoxy-based spherical SPMBs to fabricate dimers, trimers, tetramers, and so on, all consisting of similar spherical particles, in order to demonstrate how they induce director structures corresponding to various higher-order elastic multipoles. SPMBs define planar boundary conditions for LC molecules as revealed by polarizing optical microscopy textures (Fig. 7a, b). Embedded magnetic nanoparticles allow for a facile control of SPMBs with a magnetic field. To form composite linear particles consisting of two or more SPMBs, individual SPMBs were placed nearby each other using laser tweezers and LC was locally melted, again with tweezers, to the isotropic state. While in the melted isotropic area, particles where arranged into the linear chains of two or more touching particles using laser tweezers, and then chains were aligned parallel to the cell's rubbing direction using holonomic magnetic control (Fig. 7c, d). They irreversibly formed dimers, trimers, or tetramers. After the locally melted LC was quenched from isotropic back to the nematic state, the magnetic field holding particle chains was removed, leaving chains stable and oriented along $\mathbf{n}_0$. The alternation of director tilt at particle surfaces with respect to $\mathbf{n}_0$ revealed by different modes of microscopy is consistent with that in the ansatzes of sources of high-order elastic multipoles (compare Fig. 7h, l and Fig. 1). We also employed the polarimetric imaging (Methods) of director distortions around colloidal particles (Fig. 7m–o), results of which were consistent with polarizing micrographs and numerically calculated director structures. While a single SPMB induces an elastic quadrupole, a pair of SPMBs in a chain oriented along $\mathbf{n}_0$ has a strong elastic hexadecapole moment (compare Figs. 7h, and 1 for $l = 4$), which can be the strongest leading-order elastic multipole for certain parameters. Similarly, a chain of three particles can have a strongly pronounced 64-pole (compare Figs. 7l, and 1 for $l = 6$).

High-order multipoles can be also achieved by chains of irreversibly bound spherical particles with conic or homeotropic anchoring on their surfaces. We use numerical calculations to probe the geometric and boundary conditions parameter space for such particles (Fig. 8) and the ensuing strengths of elastic multipole moments. Defect configurations and $\mathbf{n}(\mathbf{r})$ structures depend on the conic anchoring angle. Within the composite particles, each homeotropic sphere generates a Saturn ring defect around its equator (Fig. 8a). However, when $\alpha$ is diminished, defects arise at the neck of the particle and at each free pole (Fig. 8a). With decreasing $\alpha$ further, obtaining tangential anchoring, a Saturn ring vanishes and only neck defects and boojums remain. In the language of elastic multipoles this means that the single spherical particle transitions from negative quadrupole (i.e., with a negative multipole coefficient $q_{21}^x$) to $2 \times 4 \times 2 = 16$ pole and then to the positive quadrupole[31]. A chain-like particle comprising two spheres transitions from a hexadecapole with a negative multipole moment ($4 \times 4 = 16$) for homeotropic anchoring to $2 \times 4 \times 4 \times 2 = 64$-pole and then to a hexadecapole with a positive-multipole moment ($2 \times 4 \times 2 = 16$) for tangential anchoring. Chain particles from three spheres could induce strong $-64$-pole, 256-pole, and $+64$-pole, while the particle comprised of four spheres can effectively act as $-256$-pole, 1024-pole, and $+256$-pole, depending on boundary conditions. To summarize, with parameters for the desired leading-order multipoles optimized, colloidal oligomer chain particles with $N$ similar spheres can act as $-4^N$-pole in the case of

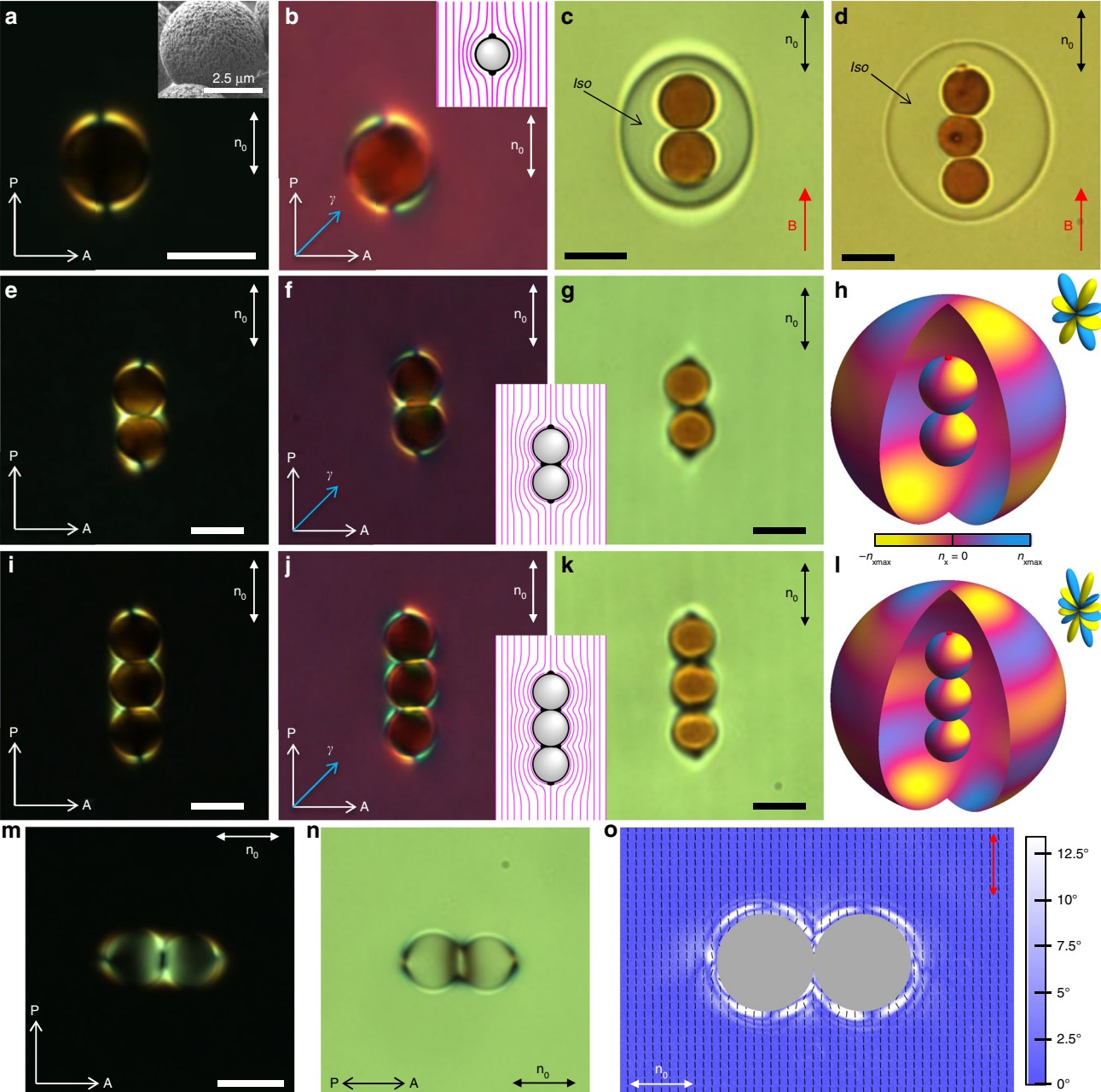

**Fig. 7** Elastic multipoles around a chain of similar particles with planar anchoring. **a–g**, **i–k** Textures of particles in a nematic cell from optical polarizing, (**b**, **f**, **j**) with and (**a**, **e**, **i**) without a retardation plate, and (**c**, **d**, **g**, **k**) bright field microscopy. P and A are crossed polarizer and analyzer and $\gamma$ is a slow axis of a full wave (530 nm) retardation plate. B is magnetic field. Inset in (**a**) shows the SEM image of the epoxy particles. Insets show a schematic diagram of the director field $\mathbf{n}(\mathbf{r})$ (magenta lines) around epoxy particles. **h**, **l** Color-coded diagrams of the $n_x$ component of $\mathbf{n}(\mathbf{r})$, which is caused by its tilt away from $\mathbf{n}_0$, directly at the surface of the particles and at the surface of the interpolation sphere. **o** Polarimetric micrograph of spatial distribution of polarization state of the green imaging light passing through the sample shown in (**m**) and (**n**) and polarized in the direction shown by a red arrow. Thin black bars indicate the orientation of the polarization ellipses and background color shows the distribution of their ellipticity according to the color-coded bar on the right. Scale bar: 5 μm

homeotropic anchoring, $2 \times 4^N \times 2$-pole for moderate conic anchoring and as $2 \times 4^{N-1} \times 2$-pole when anchoring is tangential. We note that neck defects at moderate $\alpha$ are inconsequential for defining the strongest leading-order multipoles, because they cause only weak or no deformations, though they can affect the strength of higher-order multipoles.

Graphs in Fig. 8b show spherical multipole coefficients $q_{l1}^x$ for chain composite colloids with various $\alpha$. These coefficients were calculated such that the center of the interpolation sphere coincides with geometrical center of the symmetric chain

composite colloids. Generally, particles show no or weak dipolar moment, except for homeotropic particle with four spheres, for which Saturn rings move upwards towards the necks and pole (Fig. 8a). Multipole moments with odd $l$ tend to be zero due to particle (and consequently $\mathbf{n}(\mathbf{r})$) symmetry (Fig. 8b). Particles with only one sphere tend to show the strongest quadrupole moment $q_{21}^x$. A sphere with homeotropic anchoring forms an elastic quadrupole with a negative magnitude, whereas the sphere with tangential anchoring is an elastic quadrupole with a positive magnitude (Fig. 8b). Interestingly, chain colloids of arbitrary

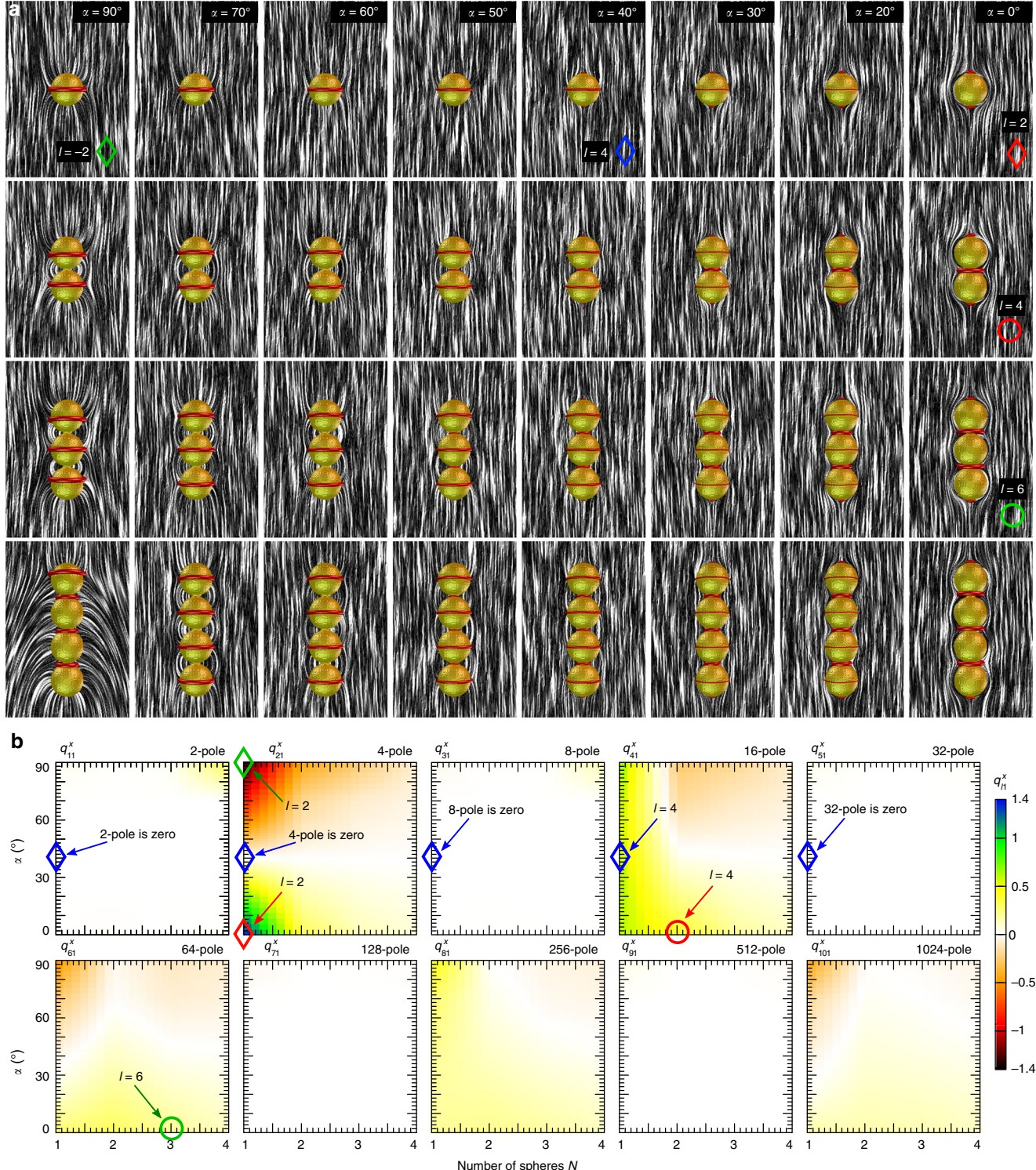

**Fig. 8** Chain particles with conic anchoring and their elastic multipole moments. **a** The defects transitions from Saturn ring on the equator (homeotropic anchoring) to combination of both Saturn rings, neck defects and boojums (moderate angle $\alpha$) and ultimately to combination of neck defects and boojums (planar anchoring). The surface anchoring tilt angle notably affects $\mathbf{n}(\mathbf{r})$-distortions around the particle and consequently its multipole moment. **b** Chain composites have mirror symmetry along $z$-axis, therefore multipole moments with odd $l$ are not present. **a** Circles show configurations and **b** parameter space also realized in the experiments (see Fig. 7). Rhombs show **a** configurations and **b** parameter space with pure multipoles. Strong quadrupole emerges for spheres with tangential or homeotropic anchoring on the surface (marked respectively by red and green rhombs). Sphere with conic anchoring angle $\alpha = 40°$ express dominant hexadecapole (marked by a blue rhomb) since the quadrupole and other multipoles are absent. The hexadecapolar moment is also dominant in particles comprising of two spheres with tangential anchoring (marked by a red circle). 64-Pole is weak, but for the particle comprising of three spheres it is the most prominent multipole (marked by a green circle)

number of spheres with $\alpha = 40°$ show no quadrupole moment. Colloidal particles with one sphere exhibit a strong positive hexadecapole moment $q_{41}^x$, which extends to multiple sphere particles, with conic anchoring angle below $\alpha = 40°$. For example, a blue rhomb (Fig. 8a) marks one particle with $\alpha = 40°$; one can see that the corresponding hexadecapolar moment is at maximum while all the other elastic multipoles lower than the hexadecapole are not present (Fig. 8b). When the anchoring angle exceeds $\alpha = 40°$, particles with multiple spheres transition into having negative hexadecapole moments. Chain composite particles can have both positive and negative 64-pole moment, whereas structures in between have none. It appears to be the strongest for spherical particles with large $\alpha$ and composites of two or three spheres with low $\alpha$, as in the composite particles observed in experiments (Fig. 7). Both 256- and 1024-pole ($q_{81}^x$ and $q_{10\,1}^x$) appear to be small, though reaching both positive and negative values as the geometric and surface anchoring parameters are varied. An interesting observation is that a sphere with homeotropic or tangential anchoring can act as strong elastic quadrupoles with opposite signs of moments. Conversely, spheres with conic anchoring at $\alpha = 40°$ exhibit a pure hexadecapole due to the superposition of opposite-charged quadrupole moments. 64-polar contribution is not significantly strong in general, however it is a prominently expressed multipole for a particle consisting of two spheres with $\alpha = 40°$.

Presented results for composite chain particles with up to two spheres are in a good agreement with predictions posed on the basis of arbitrary multipole creation principle discussed above (Eq. (5)). However, colloidal composites comprising three and especially four spheres are more complex. The apparent mismatch arises because the particles are long with respect to the typical range of deformation in the nematic LC, objects extended along $\mathbf{n}_0$. Consequently, the corresponding distortions in $\mathbf{n}(\mathbf{r})$ do not resemble pure analytic multipoles (Fig. 1), but rather decompose into large number of multipoles. The problem could be mitigated by using oblate spheroids instead of spheres as building blocks of the chain colloids, putting the induced topological defects closer together, so that the alternation of director tilt could more closely mimic the corresponding ansatzes (Fig. 1).

## Discussion

Our experimental results and numerical calculations show that one can design colloidal particles with different elastic multipoles, including the higher-order ones, by changing the shape and boundary conditions of constituent particles. For example, using dimer particles with different size and surface anchoring of constituent lobes (Fig. 3), we could design colloidal particles with enhanced 64-pole (a green circle in Fig. 6). Numerically calculated director fields (Fig. 5) and diagram (Fig. 6) for strength of multipole moments of such dimer particles provide insights for designing colloids with enhanced desired multipoles, showing how strongly pronounced multipoles of different order can be preselected by varying boundary conditions and particle geometry. Following a similar strategy, we also designed a colloidal dimer with octupolar-like configuration of $\mathbf{n}(\mathbf{r})$ (Fig. 4) and the enhanced elastic octupolar moment (a blue circle in Fig. 6). It is interesting that the same arrangement of two constituent particles can also allow for defining $\mathbf{n}(\mathbf{r})$ with enhanced dipolar moment (a red circle in Fig. 6) under different conditions (Fig. 4d, h). The calculated multipolar moment diagram includes also configurations with dominant or "pure" elastic multipoles as quadrupolar, octupolar and hexadecapolar (with the latter marked by a green diamond in Fig. 6). As one can see from a diagram, the hexadecapolar elastic moment is pronounced to a smaller or larger extent in all configurations, and that the conditions for it to be a leading-order multipole can be created in multiple ways.

Composite particles with different higher order elastic multipoles can be also designed using colloidal oligomers formed by similar connected spheres (Fig. 7), which is consistent with the results of numerical calculations (Fig. 8). For example, a dimer of two particles with tangential surface anchoring, which each separately have a leading quadrupolar elastic moment, together give a rise to strongly enhanced hexadecapolar moment (marked by a red circle in Fig. 8). Three particles show the configuration of $\mathbf{n}(\mathbf{r})$ with the director tilt reversals characteristic for a 64-pole with detectable corresponding elastic moment (Fig. 1).

The presented strategies allow for the design of colloids with a variety of elastic multipoles, including high leading-order multipoles like hexadecapoles and also "mixed" multipoles with strongly pronounced multipoles of different order. While having "pure" elastic multipoles is fundamentally interesting and has the advantage that they can be used for designing colloidal self-assemblies on the basis of corresponding well known interaction potentials, nematic colloids with "mixed" multipoles can cover even larger diversity of anisotropic elastic interactions (Fig. 3). The strategies described in this work, which involve dimers, trimers, and oligomers of similar or dissimilar colloidal spheres, are just examples showing how the uniform alignment of the nematic host can be locally perturbed to mimic the corresponding ansatzes of elastic multipoles (Fig. 1). However, similar multipolar director distortions can be also achieved using colloidal objects with complex shapes obtained by means of photolithography[18,21,42] and two-photon-polymerization[43,44]. On the other hand, the concept of controlling surface anchoring on spherical constituents of composite colloidal objects that we present here can be extended to patchy particles[7–9,45], where different patches can exhibit different boundary conditions, and particles with controlled surface topography[46], surface charging[47], and chemical functionalization[15]. In nematic hosts, these highly tunable multipolar elastic interactions can be further enriched by weakly screened electrostatic monopole-like[48] and magnetic dipolar[49,50] interactions in cases of charged or magnetic particles. The ability of describing elastic, electrostatic, and magnetic interactions as multipoles of different nature and order is a useful platform for designing LC colloidal composites. Interestingly, in this respect the electrostatic monopoles in LCs have been studied[48], but designing higher order electrostatic multipoles appears to be challenging so far. Differently, magnetic monopoles are considered impossible while dipoles can be easily obtained by using magnetically monodomain particles[49,50]. Elastic dipoles, quadrupoles, and hexadecapoles have been studied previously[15], and now the spectrum of accessible elastic multipoles is significantly broadened by this work. We envisage that this "zoo" of multipoles of different nature and order, which now by far exceeds the diversity of chemical elements similarly described by use of spherical harmonics[15], will be useful in "on demand" designing and realizing composite materials with desired structure and composition. In the case when nematic colloids have mixed multipoles with comparable strengths, although lower-order multipoles within them will define the behavior at large distance due to the inverse power type of scaling of elastic potential, the higher order multipoles can still significantly influence the behavior of elastic colloids at short center-to-center distances, which is where the details of self-assembled colloidal superstructures are defined. Although the confinement of LC colloids into thin glass cells with strong or finite boundary conditions, these confinement effects are expected to influence the range and strength of colloidal interactions again similar to that between electrostatic charge distributions in proximity of surfaces with various charging and boundary conditions, which

again could be modeled by invoking elastic analogs of image charges[35,36].

Elastic multipoles are in general tensorial quantites, which might lead to interesting approaches for their hybridization, analogously to hybridization of atomic orbitals. The colloidal particles used in this work are rotationally symmetric (along the z-axis) and also impose rotationally symmetric anchoring profile, which results in the fact that all multipole coefficients of these particles are scalars (i.e., single numbers). However, in full definition, elastic multipoles (coefficients) are tensorial quantities with tensor rank related to the order of the multipoles[26]. Such generalized elastic multipoles can be realized by breaking symmetries of the nematic distortion field around colloidal particles, for example by designing particle geometry or imposed surface anchoring pattern (i.e., patchy particles). Thus, if having controllably realized such tensorial elastic multipoles, one could open routes for hybridization of multipoles, directly affecting also the interparticle interactions and self-assembly.

While it is natural to think about building of the nematic colloid and atom analogy and pursuing colloidal assemblies that resemble small molecules, polymers, and crystals, the high-order elastic multipoles do not have direct analogs in atomic systems. Since one characteristic feature that they exhibit is the large number of alternating attractive-repulsive sectors in the pair interaction potentials, we foresee that self-assembly of such particles may result in new forms of the colloidal analogs of spin glasses, where a large variety of structures with comparable energies can occur[51]. For example, at a given center-to-center distance, the pair interaction energy between two colloidal 64-poles (Fig. 3) has 12 local/global minima corresponding to the sectors of attraction, whereas only four nearest-neighbor sites can be occupied in a two-dimensional plane simultaneously, giving the origins to multiple accessible states of the colloidal system, all with comparable energies. This rich energetic landscape may allow for forming colloidal assemblies with multiple states of comparable energy, as in the colloidal spin ice systems[51] in both two and three dimensions. Many-body and kinetic effects may become important in these defining assemblies and physical behavior and will be of great interest to explore in future studies.

Methodologically, our work is based on a combination of experiments and numerical modeling, where experimentally wet chemistry is used to variably create gourd-shaped particles and a combination of optical microscopy techniques is used to determine their multipolar properties, whereas numerical modeling is based on phenomenological free energy minimization approach to calculate the ordering fields, and is then complemented by multipolar expansion algorithm into spherical harmonics. This approach can be effectively extended to other potential strategies of designing elastic multipoles discussed above, as well as supplemented by adding magnetic and electrostatic interactions[48–50]. Since the colloidal objects can have different compositions, including constituents made of noble metals[49], magnetic materials[46,49,50,52], semiconductor nanoparticles[47,48], and dielectric objects[17,18,21,24,31,42–44] (with means of defining boundary conditions for $\mathbf{n}(\mathbf{r})$ on such colloidal objects already demonstrated[17,18,21,24,31,42–50,52,53]), we envisage that properties of the ensuing colloidal composite metamaterials can be pre-engineered by expanding the above described design toolkit to account for collective behavior of such assemblies enriched by plasmonic resonances[32–34], plasmon–exciton interactions[54,55], etc.

To conclude, this work demonstrates realization of colloidal atoms from high-order multipoles based on geometrical and topological design of distortion fields in nematic colloidal fluids. The high-order multipolar colloidal objects are realized from elastic multipoles in the orientational director fields of nematic fluid that also can transfer interparticle interactions of multipolar

symmetry. We show realization of colloids with dipolar, quadrupolar, octupolar, hexadecapolar, 32-polar, and even 64-polar multipole components, that we show not only can be controllably varied, but also designed by controlling particle shape and surface anchoring boundary conditions. Interestingly, we are also able to identify regimes, i.e., colloids with distinct geometrical and surface parameters, of "pure" or leading-order multipoles, where a single-multipole dominates and leads the structure, and "non-pure" or "mixed" multipoles, where various combinations of different multipoles are present on a single particle and determine the system.

More generally, this work is a contribution towards developing a novel, colloidal, matter that rather uniquely can go beyond the interaction types that are possible in the set of known atoms as determined by their orbitals. We show design of high-order multipoles, such as hexadecapole, 32-pole and even 64-pole, that can be mapped to atomic orbitals (subshells) of $l = 4$–6, respectively, which do not have direct analogs in atomic and molecular systems. At this stage, our work is primarily centered around demonstrating the capabilities to realize individual particles—colloidal atoms, and basic interactions. In this work, the interaction range of high-order multipoles is also limited by relatively short range decay of high-order multipoles, but we believe that it is exactly by combining geometrical and topological approaches, that one could possibly open a field to beyond-atomic matter with novel material properties.

Our work shows that, similar to how we often think about high-order multipolar charge distributions in electrostatics, where high-order multipoles emerge from superposition of the lower-order ones when lower-order multipoles mutually cancel, high-order elastic multipoles can be designed by superimposing the lower-order ones and tuning conditions for cancelation of multipoles up to the desired leading-order one. The illustrative examples of this are composite colloidal particles, where each of them individually would induce a lower-order multipole, dimers, trimers, tetramers, and oligomers of such particles can prompt creation of high-order multipoles under proper conditions (e.g., a dimer of colloidal quadrupoles can be arranged so that all lower-order multipoles but hexadecapolar cancel, making an elastic hexadecapole). These insights offer simple but powerful means for designing self-assembled colloidal composites.

## Methods

**Materials**. We used a room temperature nematic LC 4-cyano-4′-pentylbiphenyl (5CB, from Frinton Laboratories, Inc.) or nematic mixture E7 (EM Industries) as a colloidal host medium. To define localized director distortions in LC, mimicking symmetry of point sources of various multipoles (Fig. 1), we used a series of different colloidal particles. Gourd-shaped polystyrene dimer particles with two lobes of different diameter $2r_a \approx 2.5$ μm and $2r_b \approx 1.25$ μm (Figs. 2, 3) were synthesized using a modified seeded polymerization technique[39–41] and first dispersed in ethanol before introducing them into the nematic host. LC molecules aligned tangentially at the surface of the small lobe and exhibited a conic alignment at the surface of the large lobe. To have a tangential alignment of LC molecules at the particle's surface, we used SPMBs (Dynabead M450, Invitrogen) with a nominal diameter of ≈4.5 μm, which contained ferromagnetic nanoparticles embedded into a highly cross-linked epoxy[46,52]. Alternatively, polystyrene spheres DC-05 (Thermo Fisher Scientific, Inc.) with a diameter of ≈5.3 μm also exhibited tangential boundary conditions. Glass particles (Thermo Fisher Scientific, Inc.) with a diameter of ≈5.1 μm treated with an aqueous solution (0.05 wt%) of N,N-dimethyl-N-octadecyl-3-aminopropyl-trimethoxysilyl chloride (DMOAP) exhibited perpendicular surface anchoring boundary conditions. All colloidal particles were dispersed in a LC host either via mechanical mixing or solvent exchange, producing dilute colloidal dispersions. After ~5 min sonication to break apart pre-existing aggregates, these colloidal dispersions in the LC state were filled in-between two glass plates spaced by glass spacers setting the gap thickness $d \approx 15$–60 μm. Planar surface boundary conditions at confining substrates were set by unidirectionally rubbed thin films of spin-coated and cross-linked polyimide PI2555 (HD Micro-System)[31]. The polar surface anchoring energy coefficient was estimated to be ~$10^{-4}$ J m$^{-2}$ for both the confining substrates and colloidal particles immersed within the LC, defining the strong boundary conditions on the corresponding surfaces. To minimize spherical aberrations in experiments involving high

numerical aperture (NA) immersion oil objectives, one of the used cell substrates was 0.15–0.17 mm thick, as needed for high-resolution imaging.

**Experimental techniques**. An experimental setup assembled around an inverted Olympus IX81 microscope was used for optical bright-field and polarizing microscopy observations with a 100× (NA = 1.4) oil objective. To study distortions of the director field caused in a uniform nematic LC background by colloids, we also utilized a polarimetric imaging setup integrated with the same optical microscope. Optical manipulations and assisted assembly of colloidal particles were realized with a holographic optical trapping system[56,57] operating at a wavelength of λ = 1064 nm and integrated with our optical microscope. Rotational manipulation of magnetically functionalized colloids was achieved using an in-house custom built holonomic magnetic manipulation system integrated within the same setup[46]. Translational and rotational motion of colloidal particles was recorded using a charge-coupled device (CCD) camera (Flea, PointGrey) at a rate of 15 frames per second and the exact spatial positions and orientations of colloidal particles as a function of time were then determined from captured sequences of images using motion tracking plugins of the ImageJ (National Institute of Health) analyzing software.

**Numerical modeling procedures**. Elastic multipoles in our study were formed using dimers or assembly of spherical particles with the same or different anchoring and dimensions. In numerical modeling of gourd-shaped colloidal particles comprised of two dissimilar spheres (Fig. 2), the radius $r_a$ and position $d_a$ of the lower larger sphere were kept constant, whereby the radius $r_b$ and position $d_b$ of the upper smaller or equal sphere were varied to achieve different structures. The anchoring on spheres was defined as strong and planar degenerate on the upper sphere and as conic degenerate on the lower sphere (Fig. 2). Both planar degenerate and conic degenerate anchoring impose distortions of $\mathbf{n}(\mathbf{r})$, which in the studied equilibrium structures are rotationally symmetric with respect to the z-axis.

The total free energy $F$ was minimized numerically by using an explicit Euler relaxation finite difference scheme on a cubic mesh[58]. Material parameters of typical nematic LCs were used in the calculations[58]: $L = 4 \times 10^{-11}$ N, $A = -0.172 \times 10^6$ J m$^{-3}$, $B = -2.12 \times 10^6$ J m$^{-3}$, $C = 1.73 \times 10^6$ J m$^{-3}$. Simulations were performed on a square grid consisting of $400 \times 400 \times 400$ simulation points. For gourd-shaped dimers, composite colloids consist of two spheres, the bottom one has 100 points in diameter, whereas the upper sphere has from 20 to 100 points in diameter and is gradually moved upwards in steps of 10 points to mimic a broad range of shapes of dimer particles that can be obtained in experiments[39–41]. The composite colloids consisting of different number of equally sized spheres have 50 points in diameter. We assume fixed homeotropic anchoring on the cell surfaces and strong conic, homeotropic or planar degenerate anchoring on the composite colloids. The strong boundary conditions were chosen to achieve optimal matching between calculated and measured polarization micrographs and the corresponding director structures.

Multipole expansion was numerically performed with Gauss–Legendre algorithm, which was implemented via the numerical library SHTns[59,60]. Several optimizations were used to achieve maximum efficiency[59], including the fast Fourier transform from the library Fastest Fourier transform in the West[61,62] to improve accuracy and speed. The main advantage of this library is the efficient on-the-fly computation of the Legendre-associated functions. Also, the algorithms implemented in SHTns are of high order accuracy $O(N^3)$, where $N$ is the number of calculation points[59].

An important method for studies of nematic structures is the Landau-de Gennes (LdG) free energy approach[38]. It is based on the full tensorial order parameter $Q_{ij}$, which incorporates the orientation of the director $\mathbf{n}$, orientation of the possible biaxial ordering relative to the director, scalar degree of order $S$ and biaxiality $P$. LdG modeling is a phenomenological approach which uses a tensor order parameter to construct a free energy functional $F$, which is also able to fully characterize the defect regions. We use one elastic constant approximation for the LdG free energy, which reads

$$F = \int_{LC}\left\{\frac{A}{2}Q_{ij}Q_{ji} + \frac{B}{3}Q_{ij}Q_{jk}Q_{ki} + \frac{C}{4}\left(Q_{ij}Q_{ji}\right)^2\right\}dV + \int_{LC}\left\{\frac{L}{2}\frac{\partial Q_{ij}}{\partial x_k}\frac{\partial Q_{ij}}{\partial x_k}\right\}dV + \int_{Surf}\left\{\frac{W}{2}\left(Q_{ij} - Q_{ij}^{(0)}\right)\left(Q_{ij} - Q_{ij}^{(0)}\right)\right\}dS,$$

(6)

where LC denotes the integration over the bulk of LC and Surf over the surface of colloidal particles. The first-term accounts for the variation of the nematic degree of order; $A$, $B$, and $C$ are material parameters. The second-term penalizes elastic distortions in the nematic state, where $L$ is the elastic constant. The final term in $F$ is surface free energy, which accounts for the LC interaction with the surface of the colloidal particle, where $W$ is the anchoring strength and we assume anchoring along preferred direction imposed by the leading eigen-pair of $Q_{ij}^{(0)}$ (i.e., with largest eigenvalue)[63]. The preferred direction of anchoring is set according to the anchoring type; note that the surface free energy in Eq. (6) imposes uniform anchoring along some distinct direction bot not degenerate (such as degenerate planar or conic). Nevertheless, in the work shown, the experimentally realized multipolar particles always exhibited rotational symmetry about the undistorted $\mathbf{n}_0$, which makes the use of such uniform surface free energy appropriate and sufficient.

**Nematic elastic multipoles**. Nematic elastic multipoles are commonly known today and used in the literature as approximations for elastic distortion profiles of nematic orientational fields that surround colloidal particles. Nematic orientational fields can be calculated analytically only for selected, typically rather simple systems; however, in most cases the general solution cannot be obtained. In colloids, the key problem usually arises in the proximity of particle surfaces, where strong spatial gradients emerge in the nematic orientational fields, which is though different to typically small gradients away from particles. Therefore, to obtain analytical insight into nematic fields, the full Euler–Lagrange equations were simplified under selected assumptions (linearized) to be analytically solvable, eventually in the far-field in terms of elastic multipoles[22,26,28,64].

The expansion to nematic elastic multipoles relies on the crucial assumption of roughly uniform director field $\mathbf{n}(\mathbf{r}) \approx (n_x, n_y, 1)$, with small $n_x, n_y \ll 1$, where note that by definition, $\mathbf{n}$, is to be a unit vector field. Typically, such assumption can be justified at sufficiently large distances from colloidal particles (such as order of magnitude one particle radius away from the particle surface or can be even less). An additional assumption is also that the nematic elastic modes (i.e. elastic free energy) are described with one single elastic constant. Taking such approximations, the full nematic elastic free energy can be simplified to the harmonic free energy $f_E = \frac{1}{2}K\sum_{\mu=x,y}\left(\nabla n_\mu\right)^2$, where we use notation $n_\mu$ ($\mu = x, y$) for components perpendicular to far field direction. The corresponding Euler–Lagrange equations are Laplace equations: $\nabla^2 n_\mu = 0$. The solution of Laplace equations is now sought in terms of series of multipoles.

The elastic multipoles can be introduced via spherical multipole moments or by using Green function[22], where the two approaches can be directly mapped one into another. For our work and analysis of results, it is convenient to use spherical harmonics as they can be readily determined by an expansion of the nematic director on a sphere that encloses the considered multipolar colloidal particles. Laplace equations for $n_\mu$ are separable in spherical coordinates and can be analytically solved, with their general solutions written as a sum of multipolar contributions $n_\mu(r, \theta, \phi) = \sum_{l=0}^{\infty}\sum_{m=-l}^{+l} q_{lm}^\mu \frac{R_{eff}^{l+1}}{r^{l+1}} Y_l^m(\theta, \phi)$, where $\theta$ is polar and $\varphi$ azimuth angle, $q_{lm}^\mu$ are spherical multipole moments coefficients and $Y_l^m(\theta, \varphi)$ are spherical harmonics. In order to extract distinct coefficient of selected multipole moment the orthogonality of spherical harmonics is used $\int_0^{2\pi}\int_0^\pi Y_l^m(\theta, \varphi)Y_j^k(\theta, \varphi)\sin\theta d\theta d\varphi = \delta_{lj}\delta_{mk}$, where $\delta_{ij}$ is the Kronecker symbol. Note that the radius $R_{eff}$ of the sphere at which the expansion is performed can be easily taken large enough to satisfy the assumptions $n_x, n_y \ll 1$ and $n_z \approx 1$.

In homogeneous background field ($\mathbf{n}_0 = \{0, 0, 1\}$), the symmetry of elastic multipolar distortions generated by particles is determined by the symmetry of colloidal particles and their anchoring. In this work we have considered only particles invariant with respect to rotations about z-axis, which have no azimuthal contribution to $\mathbf{n}(\mathbf{r})$, hence the director field imposed by a selected particle should be invariant with respect to rotations about $\mathbf{n}_0$. This constraint sets monopole coefficient $A^\mu = 0$ and furthermore, implies that $q_{lm}^\mu$, with $m = \pm 1$ are the only nonvanishing coefficients in the expansion (Eq. (4)) of the Cartesian director field components $n_\mu$.

The Laplace equation has no inherent length scale; therefore, also no inherent length scale is present in the multipolar expansion as solution of the Laplace equation for the nematic director components. Nevertheless, clearly, already from the perspective of the dimensional analysis, multipoles have units of powers of a certain length scale. In our case of nematic elastic multipoles, we introduce this certain length scale as $R_{eff}$ (e.g., as introduced in Eq. (4)), thus making the elastic multipolar coefficients $q_{lm}^\mu$ dimensionless to allow for comparison of the magnitudes of different multipoles. In colloidal systems, this certain length scale is naturally related to the particle size, which for spherical particles is the radius. However, for more complex shaped particles, like our gourd particles, it is less clear how to select this scale $R_{eff}$, especially if wanting to effectively compare multipole coefficients of particles of somewhat different shapes. Therefore, some selection has to be made according to the leading geometrical elements (shape) of the particles.

**Calculation of spherical multipole coefficients**. The calculation of spherical multipole coefficients (Eq. (4)) is performed with forward spherical harmonic transformation[59,60]. The calculations were performed numerically using numerical library (SHTns), in two consecutive steps. In the first step, the integral over $\varphi$ is performed by calculating the Fourier transform

$$q_m^\mu(\theta) = \int_0^{2\pi} n_\mu(\theta, \varphi)e^{-im\varphi}d\varphi,$$

(7)

and in the second step we calculate the Legendre transform

$$q_{lm}^\mu = \frac{r^{l+1}}{R_{eff}^{l+1}}\int_0^\pi q_m^\mu(\theta)P_l^m(\cos\theta)\sin\theta d\theta.$$

(8)

The SHTns library uses for the forward spherical harmonic transform data written in spherical coordinates on a sphere, specifically, on nodes of a sphere with discretized latitude $\theta_i$ and longitude $\varphi_i$, which are equally spaced along longitudinal coordinate and Gaussian along the latitude. Such distribution of numerical nodes

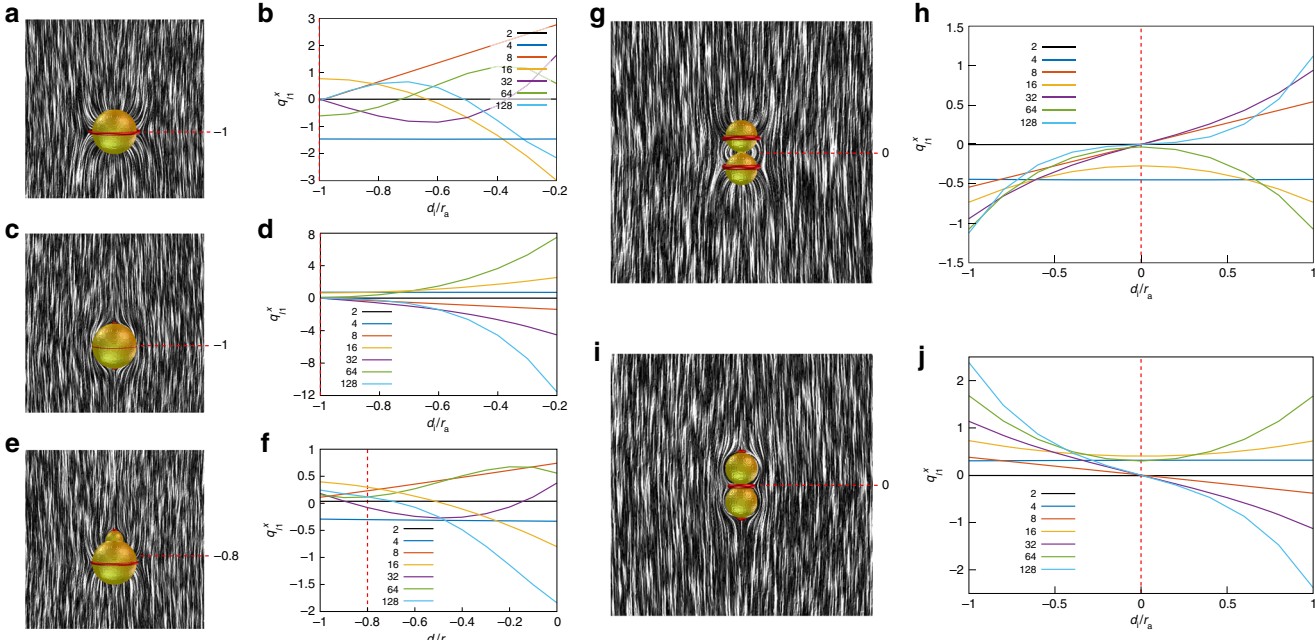

**Fig. 9** Examples of spherical multipole moments of composite nematic colloids. The value of $q_{l1}^x$ was calculated for selected simulated composite nematic colloids and chain particle comprising of two equal spheres. **a** Colloidal particle with homeotropic boundary conditions centered at $r_a$ below the origin induces a quadrupolar distortion of $\mathbf{n}(\mathbf{r})$, commonly known as a Saturn ring configuration. **b** Corresponding graph shows constant quadrupole coefficient at all displacements $d_i$, other multipoles have extreme or zero at $d_i = -r_a$, corresponding to the geometrical center of the spherical colloid. **c** Spherical colloidal particle with conic anchoring at angle $\alpha = 20°$ centered at $r_a$ below the origin. **d** Plot shows that quadrupolar coefficient is constant at all $d_i$, whereas other coefficients higher then 16-pole are zero when center of the interpolation sphere coincides with geometrical center $d_i = d_a$. **e** Composite colloidal particle at $r_a$ below the origin with $d_b = 0$, $r_b = 2r_a/5$ and conic anchoring at angle $\alpha = 60°$. **f** Quadrupolar coefficient is constant regardless the position of the interaction sphere, whereas higher multipole moments show complex variations even at the geometrical center. Geometrical centers of the composite colloids are depicted with red dashed line. **g** Chain colloid with homeotropic anchoring on the surfaces induces two Saturn rings. **h** The director field shows strong quadrupolar and hexadecapolar, and weak 64-polar contribution. Higher multipoles are zero in the geometrical center of a composite chain colloid. **i** Chain colloid with tangential anchoring induces strong neck defect and two boojums at the poles. **j** The director field shows strong hexadecapole, quadrupole and 64-pole, whereas higher multipoles are zero in the geometrical center. The location of the geometrical center of the colloid is depicted with red dashed line (at $d_i = 0$)

(points) gives more balanced representation of a waist region of the sphere in comparison to the poles as if using, e.g., regular grids.

The director field $n_\mu$ used for determining spherical multipole coefficients is obtained from numerical calculations based on free energy minimization and is calculated on discrete points of a cubic mesh, which do not match with points on the sphere. Therefore, we perform trilinear interpolation of the tensor order parameter $Q_{ij}$, to get $Q_{ij}$ at arbitrary point in space

$$
\begin{aligned}
Q_{ij}(x, y, z) = &[1-x][1-y][1-z]Q_{ij}(0,0,0) + x[1-y][1-z]Q_{ij}(1,0,0) \\
&+[1-x]y[1-z]Q_{ij}(0,1,0) + xy[1-z]Q_{ij}(1,1,0) \\
&+[1-x][1-y]zQ_{ij}(0,0,1) + x[1-y]zQ_{ij}(1,0,1) \\
&+[1-x]yzQ_{ij}(0,1,1) + xyzQ_{ij}(1,1,1).
\end{aligned}
\tag{9}
$$

where $(x, y, z)$ with $x, y, z \in [0,1]$ denotes a location within a selected cube of eight neighboring points of a square lattice; each point is in the corner of the cube and the corners are labeled with vector of 0 and 1.

Performing the calculation of spherical multipole coefficients on the discrete grid, the integral (Eq. (7)) reduces to the discrete Fourier transform and the use of the Gauss–Legendre quadrature replaces the integral (Eq. (8)) with the sum

$$
q_{lm}^\mu = \frac{r^{l+1}}{R_{eff}^{l+1}} \sum_{j=1}^{N_\theta} q_m^\mu(\theta_j) P_l^m(\cos\theta_j) w_j,
\tag{10}
$$

where $\theta_j$ and $w_j$ are Gauss node angles and Gauss node weights, respectively, and $N_\theta$ is the number of discrete points in latitude.

The radius of the interpolation sphere must be chosen such that the distortions from homogeneous alignment $n_\mu$ are rather small compared to $n_z \approx 1$. We perform the multipole analysis by setting the magnitude of the maximum allowed transversal director field component to be $n_x = 0.1$ on the entire interpolation sphere (see also Fig. 2), where we determine the appropriate interpolation sphere radius $r_i$ with bisection.

For the effective size of the particles, we take that the effective radius to be half the length of the particle's dimension in $z$ direction, which can be written (using parameters from Fig. 2) as $R_{eff} = (2r_a + d_b + r_b)/2$. Note, that we tested various

possible selections for the effective radius and for the systems shown, this selection gives most reasonable results; especially, such selection of $R_{eff}$ accounts rather well for the changes in the geometry of our particles and makes the multipole coefficients comparable in magnitude. Finally, selecting the effective radius, it also defines the center of the composite colloid, which we call the geometrical center and is depicted with a red dot in Fig. 2.

Important parameter in the calculation of the multipolar coefficients is also the position of the interpolation sphere $d_i$, i.e., the actual location of the multipoles. Note that if the distortions of $\mathbf{n}(\mathbf{r})$ are symmetric up–down along $\mathbf{n}_0$ (in our case along $z$-axis), the location of the multipole is clearly at the mirror plane. Also, if the distortions are rotationally symmetric around $\mathbf{n}_0$, the location of the multipole is along the rotational axis. However, the location of the multipoles and, correspondingly, the choice of the interpolation sphere position become less clear if the distortion (and particle) are asymmetric. In this work we take the interpolation sphere to be centered in the geometric center, which we analyze in more details by varying the location of the interpolation sphere as shown in Fig. 9.

Figure 9a–f shows the spherical multipole coefficients $q_{l1}^x$ as a function of the interpolation sphere position $d_i$. As first example, we present a spherical particle with homeotropic anchoring on the surface (Fig. 9a), which has the director structure of an elastic quadrupole, commonly known as the Saturn ring configuration. Figure 9b shows that dipole moment is zero, quadrupole is constant regardless of $d_i$, whereas all higher multipoles are present nonetheless with their magnitudes dependent on $d_i$. Notably, if the center of the interpolation sphere coincides with the geometrical center of the colloidal particle, the multipole coefficients have an extreme or zero, which actually one would expect, and supports the relevance of the geometrical center as the location of the multipoles. Note that higher multipole moments emerge primarily because the particle is positioned away from the center of the simulation box (in which we calculate the $\mathbf{Q}$ tensor profile) and the confinement distorts the exact quadrupolar symmetry of $\mathbf{n}(\mathbf{r})$. As second example, a spherical particle with conic degenerate anchoring is presented (Fig. 9c, d). The quadrupole coefficient is observed to be constant for all $d_i$; however, other higher multipole coefficients emerge as well and are again dependent on the interpolation sphere position. But, interestingly, again, when the center of the interpolation sphere coincides with the geometrical center of the colloidal sphere at $d_i = -r_a$, all multipoles higher than

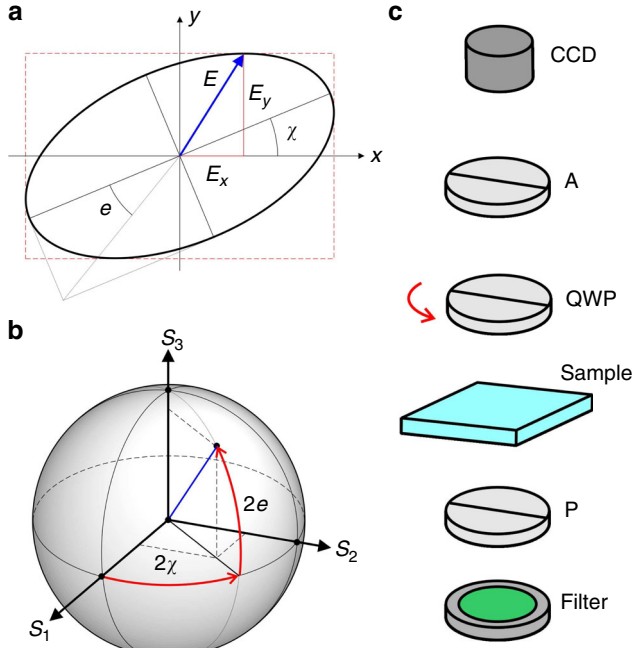

**Fig. 10** Polarimetric imaging principles and setup. **a**, **b** Polarization ellipse and a Poincaré sphere for polarized light. **c** Experimental setup for measuring orientation and ellipticity of the polarization ellipse

16-pole drop to zero. This again supports the relevance of geometric center as a reasonable position for the center of the interpolation sphere in determining the multipole coefficient. Nevertheless, less clearly, Fig. 9e, f shows the multipole coefficients of a composite colloidal particle with upper sphere radius $r_b = 2/5r_a$ and at position $d_b = 0$. The quadrupolar moment is constant over all positions of the interpolation sphere, but higher multipoles emerge as well, and notably without clear signature at the geometrical center (such as zero value or maximum/minimum). This result shows that although our use of the geometrical center of particle works well over a range of regimes of particle shapes and anchoring types, as it corresponds exactly to the location of the multipoles, in general, it is only a reasonable approximation. Another example of systems where only geometrical arguments for finding location of multipole centers will fail are (symmetric) colloids (even spheres) with different anchoring strengths (and/or anchoring types) on different parts of the colloidal surface. Overall, this indicates an interesting possible further study, such as if and how, different multipoles of one particle could possibly emerge at different mutually shifted locations.

Figure 9g–j shows two chain colloids comprising of two equal spheres joined at the poles. Figure 9g shows a pair of colloids with homeotropic anchoring. Two Saturn rings emerge at the waist/equator of each sphere, creating distinct $\mathbf{n}(\mathbf{r})$-deformations of the hexadecapole with $n_x > 0$ (see Fig. 1). Nevertheless, as shown by the full analysis (Fig. 9h), the hexadecapolar deformations are also accompanied by the quadrupolar and weak 64-pole components. Figures 9i, j demonstrates another case of a colloidal chain, now with tangential anchoring on the spheres. The nematic profile shows a defect region in the neck of the particle and boojums at each free pole. The corresponding $\mathbf{n}(\mathbf{r})$ resembles the hexadecapole $n_x < 0$ (as shown in Fig. 1), effectively, somewhat stretched along $z$-axis, which turns out excites other (symmetrical along $z$-axis) multipoles. For this case, the strongest multipole is the hexadecapole, followed by the quadrupole and 64-pole, whereas other multipoles are zero in the geometrical center of the composite chain particle, as conditioned by the symmetry of distortions.

**Polarimetric imaging of colloids in LCs**. In addition to the standard technique of polarizing optical microscopy, we used polarimetric imaging of structures around colloidal particles with measurements of parameters of polarized light emerging from the sample on a pixel-by-pixel basis. To determine the orientation $\chi$ and ellipticity $e$ of the light's polarization ellipse after traversing the nematic sample with a colloidal inclusion (Fig. 10a, b), we used the rotating quarter-wave-plate (QWP) measurements[65]. The measurement setup is shown in Fig. 10c, where we used a narrow band filter with central wavelength at 546 nm after a halogen lamp as a light source. The light incident on the sample was polarized with a linear polarizer. In the optical path, the QWP is inserted after the sample and is followed by an analyzer fixed along $x$-axis. This setup allows for the measurements of polarization ellipse parameters of light passing the sample by using intensities of light transmitted through the system polarizer-sample-QWP-analyzer at different QWP orientations. The QWP can be rotated by an angle $\theta_P$ with respect to analyzer

direction and Stokes parameters (Fig. 10b) can be found as follows[65]:

$$S_0 = Z_1 - Z_3, \; S_1 = 2Z_3, \; S_2 = 2Z_4, \; S_3 = Z_2, \qquad (11)$$

where coefficients $Z_1$, $Z_2$, $Z_3$, and $Z_4$ are given by

$$Z_1 = \frac{2}{N_p}\sum_{i=1}^{N_p} I_i, \; Z_2 = \frac{4}{N_p}\sum_{i=1}^{N_p} I_i \sin 2\theta_{pi}, \; Z_3 = \frac{4}{N_p}\sum_{i=1}^{N_p} I_i \cos 4\theta_{pi}, \; Z_4 = \frac{4}{N_p}\sum_{i=1}^{N_p} I_i \sin 4\theta_{pi}, \qquad (12)$$

where $N_p$ is a number of angels $\theta_{pi}$ at which the intensity $I_i$ of transmitted light was measured. We measured the transmitted light intensity at orientations of the QWP fast axis with respect to an analyzer from $\theta_p = 0°$ to $\theta_p = 180°$ with a step of 22.5°. Following this procedure, polarization ellipse parameters $\chi$ and $e$ can be determined from expressions $\tan 2\chi = S_2/S_1$ and $\sin 2e = S_3/S_0$. The intensity of transmitted light after an analyzer corresponding to each pixel was recorded with a CCD camera. A large matrix of intensities corresponding to pixels of camera was recorded for each $\theta_p$, and polarization parameters were calculated for each pixel, yielding polarimetric images of colloidal particles and distortions around them (Fig. 7o). As shown using the examples of dimer composite colloidal particles, the polarimetric imaging results are consistent with polarizing micrographs and numerically calculated director structures, revealing how different multipoles can be induced by studied composite colloidal objects.

## Data availability

All data are available from the authors upon reasonable request.

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

## Acknowledgements

We acknowledge technical assistance of Owen Puls and discussions with B. Fleury, M. Tasinkevych, N. Wu, and S. Čopar. Research at CU-Boulder (B.S. and I.I.S.) was supported by the U.S. Department of Energy, Office of Basic Energy Sciences, Division of Materials Sciences and Engineering, under the Grant DE-SC0019293 and by the National Science Foundation Grant DMR-1420736 (imaging and fabrication facilities). Research at FMF UL and IJS (J.A. and M.R.) was supported by Slovenian Research Agency Grants (Grant nos. J1-7300, L1-8135, and P1-0099) and US Air Force Office of Scientific Research, European Office of Aerospace Research and Development (Grant no. FA9550-15-1-0418, and Contract no. 15IOE028).

## Author contributions

B.S. and I.I.S. performed the experimental work and J.A. and M.R. performed the numerical calculations. B.S., J.A., M.R., and I.I.S. analyzed the data and wrote the paper. I.I.S. conceived and directed the project.

## Additional information

**Competing interests:** The authors declare no competing interests.

