## [Peer Review File · Nature Communications]

Reviewers' comments:

Reviewer #1 (Remarks to the Author):

I have read the article by Senyuk et al. entitled "High-order elastic multipoles as colloidal atoms". The authors performed a study to generate multipoles of higher order within nematic liquid crystals. They have supported their claims using both experiments and numerical modelling. The parameters used in this study was the size of the colloidal particles, the anchoring condition of LCs at their interfaces, the positioning of the particles with respect to each other and the number of particle clusters. At the end of the study, they have provided design rules for the multipoles that is a function of the type of the defects formed at the vicinity of the particle interfaces. The study reported in this manuscript is interesting, consistent and of high novelty that deserves publication in Nature Communications so that it can reach to their broad readership.

Below I provide some specific points about the manuscript.

- 1- The authors have used strong anchoring ($W = 10^{-2} \text{ J/m}^2$) of LCs on the surfaces of colloidal particles. Are there any measurements of this parameter experimentally for the particles used in the study?
- 2- The model in Fig. 2 may require a detailed explanation. For example, what is the physical significance of the red-dashed line represented as x in the figure? What are the significances of the distances d_i and d_b that are defined in reference to this line?
- 3- The interaction zones plotted in Fig. 3g is interesting. How does the strengths of these interactions change with position? Are there any influence of the sizes r_a and r_b of the spheres on the interaction strength profile?
- 4- Regarding Fig. 3h, in line 171, the text reads "...the experimental angular diagram (Fig. 3h) is more complicated, pointing out a richer behavior that can be understood by presence of other non-zero elastic multipoles and consistent with the lack of symmetry plane orthogonal to n_0". I found it hard to relate the plots provided in Fig. 3h with this text. A detailed explanation of what is reported in the figure and its relation to this text may help the reader.
- 5- Figures 5, 6, and 8 are complicated and sometimes hard to follow with the related text. Simplification these figures or placing some pointers regarding the features discussed in the text may help.

In addition to these, there are several typographical errors that I list below;

Line 184 – "... magnitudes of the ...",

Line 189 – SPMB abbreviation was not defined before,

Line 364 – "... forms an elastic quadrupole..."

Reviewer #3 (Remarks to the Author):

The work by Smalyukh et.al. studied the creation of elastic multipole by dispersing colloidal dimer particles into small-molecule nematic liquid crystal hosts, with the hypothesis that shapes and surface boundary conditions of the dimers would readily define the types of multipole that can be produced. Through experimental observations as well as numeric simulations, the authors described the realization of various elastic multipoles—in particular the 32- and 64-pole, which was not previously demonstrated. The investigation is systematic, providing the underlying physical principles and an effective means to design colloids with low- and high-order elastic multipoles. Analogies are also made

between the multipoles and the orbitals of atoms, which extends the importance of work to both natural and unnatural material structures.

Although the current study is solid, its significance is of some concern. Previously, the author's group has studied various types of particle shapes in order to obtain the elastic defects and interactions. This includes triangles and squares (science 2009), and knots (Nat. Mat., 2014), which demonstrated lower-order elastic dipoles and quadrupole. Most recently, the hexadecapole was reported in spherical colloids (Nat. Commun. 2016). In some cases, the self-assembly of those colloidal multipoles was demonstrated. While the current study extended the shapes to colloidal dimers with the same or different surface/composition, compared the hexadecapole previously reported, the discovery of 32- and 64-pole seems natural and not as surprising.

However, the dimer shapes selected in the current study have their own advantages, which is the continuous tunability for various parameters (diameters, center-center distance, and conic angles, etc.). This has allowed much control over the multipoles one can create since the dominating multipoles are evolving as the shape changes, as demonstrated by the simulation results. Since colloidal dimers with size/composition/aspect ratio are easily accessible through facile synthesis, the author should consider to conduct a relatively more systematic study to verify the numeric simulation.

In addition, the manuscript is somewhat lengthy, and feels like that many stories and presented in a busy manner, ranging from atomic orbitals to metamaterials and telecommunications. While some terms are there without introduction (e.g., $n(r)$, SPMB), and sometimes sentences are repeating (e.g., synthesis of dimers by van Der Waals interaction, and reasons why SPMB is brownish, etc.). Revision should be done.

Lastly, colloidal dimers (or Janus particles) are anisotropic particles widely used in the field of colloidal assembly due to directional interactions. The author may consider the demonstration of some simple assembly structures based on the elastic multipole arguments presented, and speculate on how higher order multipole may expand the assemblies of Janus dimer particles.

In summary, although the new shapes studied are desirable in the field, I have some concern about the suitability of it being published in Nat. Commun, given the fact that investigation on multiple other shapes have been reported. Some assembly structures that reflects conclusions of the study should strengthen the paper.

Reviewer #4 (Remarks to the Author):

This is an extremely interesting manuscript that describes experiments and theory that push the paradigm of "colloids as giant atoms" substantially beyond the current status-of-the-art. Of course the field has seen colloidal spheres that mutually interact with spherically symmetric potentials, dipolar interactions by magnetic cores or by high-frequency electric fields, and patchy particles and DNA-coated particles with specific binding sites and binding geometries. However, the present manuscript gives a full detailed recipe to construct multipolar colloid-colloid interactions by coupling (anchoring) the colloidal particles to the nematic director field of a molecular liquid crystal medium. As consequence, monopoles, dipoles, quadrupoles, and much higher multipoles appear, with structures that can be controlled by the anchoring conditions on the colloidal surface and the shape of the colloidal particles. This holds the prospect of a zoo of (self-assembled) structures of which the authors

only scratched the surface. The opening of such a field is worthy of being published in Nature Communications. The manuscript is well written and clear, and contains beautiful graphic material. The experiments and the theory are in good harmony, and the good agreement suggests that the system is actually very well understood in terms of nematic elasticity.

I have a few questions that popped up upon reading the manuscript. Their answer could expand the horizon of this paper even more, if not from an experimental than perhaps from a theoretical perspective:

1. The authors draw analogies with atomic orbitals, which is fascinating, but makes one wonder if there is an elastic analogy with "orbital hybridization"? In atomic systems this phenomenon takes place because of "non-diagonal" matrix elements. Could one for instance envisage a diamond-like structure by s-p³ hybridization as takes place in carbon, with possibly great consequences for photonic bandgap materials? Or likewise for e.g. graphene-like structures?

2. Atomic orbitals are purely electrostatic in nature, stemming from the $1/r$ Coulomb potential in 3D such that the potential satisfies the Laplace equation -not unlike the director field in the present study. However, in colloid science I would always expect impurities and/or surfaces that will give rise to screening; in an electrostatic analogue the length scale is the Debye length, in a superconductor the penetration depth. I guess there must be a liquid analogue as well? I suppose, given the good agreement between theory and experiment in this case, that this "screening length" is much larger than the particle size, but this probably depends on the stiffness of the LC and could perhaps also be tuned experimentally?

Perhaps the authors can comment on these issues in order to put limits on the regime of applicability of their system and ideas. Clearly, however, these answers could also be part of a research agenda that is opened-up by this work, that I recommend for publication.

Changes to Account for the Referees' Suggestions.

Report of Reviewer #1:

I have read the article by Senyuk et al. entitled "High-order elastic multipoles as colloidal atoms". The authors performed a study to generate multipoles of higher order within nematic liquid crystals. They have supported their claims using both experiments and numerical modelling. The parameters used in this study was the size of the colloidal particles, the anchoring condition of LCs at their interfaces, the positioning of the particles with respect to each other and the number of particle clusters. At the end of the study, they have provided design rules for the multipoles that is a function of the type of the defects formed at the vicinity of the particle interfaces. The study reported in this manuscript is interesting, consistent and of high novelty that deserves publication in Nature Communications so that it can reach to their broad readership.

Authors:

We thank the Referee #1 for the very positive report, describing our manuscript as "*is interesting, consistent and of high novelty.*" We are glad the Referee #1 finds that our manuscript "*deserves publication in Nature Communications so that it can reach to their broad readership*" and thank for useful comments. We have accounted for all suggestions of this report (as detailed below) that helped us to significantly improve the overall presentation of our results.

Referee #1:

Below I provide some specific points about the manuscript.

1- The authors have used strong anchoring ($W = 10^{-2}$ J/m²) of LCs on the surfaces of colloidal particles. Are there any measurements of this parameter experimentally for the particles used in the study?

Authors:

Thank you for the good question, as anchoring is one of the key material parameters. Numerically, we performed simulations with various anchoring strengths on the surfaces of the particles, from very strong ($W = 10^{-2}$ J/m²) to weak anchoring ($W = 10^{-6}$ J/m²). The simulation results were then compared with the experimental results and the simulated polarization micrographs of colloidal particles with strong anchoring matched best with the experimental polarization micrographs. We also note that the effective strength of anchoring and its role in defining the field configurations depend on the interplay of characteristic length scale (particle size versus K/W and the nematic coherence length). The experimental anchoring strength is $W=10^{-4}$ J/m² and the numerical modeling boundary conditions matched these estimates. We now clarify this in the revised manuscript.

Referee #1:

2- The model in Fig. 2 may require a detailed explanation. For example, what is the physical significance of the red-dashed line represented as x in the figure? What are the significances of the distances d_i and d_b that are defined in reference to this line?

Authors:

We thank the Referee #1 for this remark. We have now notably updated the caption of Fig 2, to more clearly explain the indicated quantities. Red and blue dashed line represent x and z axis of Cartesian coordinate system: z axis connects both centers of the spheres, whereas x axis is perpendicular with the objects and corresponding fields being rotationally symmetric. The origin of the coordinate system was fixed in the center of simulation box, whereas the position and radius of the upper sphere was varied. The position of the center of the upper sphere regarding to coordinate system is marked with d_b . The d_i sets the position of the interpolation sphere, which was in the following used for calculations of spherical harmonics. We show that the optimal position of the center of the interpolation sphere is in the geometric center of the composite particle (but in principle could be also different).

Referee #1:

3- The interaction zones plotted in Fig. 3g is interesting. How does the strengths of these interactions change with position? Are there any influence of the sizes r_a and r_b of the spheres on the interaction strength profile?

Authors:

We thank the Referee #1 for these remarks. Fig. 3g is obtained for a pure 64-pole, with all other multipole expansion coefficients equal zero. In this case, the pair interaction potential decays with distance as $1/R^{13}$, similar to how it would be the case for the interaction of electrostatic 64-poles. We note, however, that there are many other factors that can influence this behavior, including the screening effects due to surface confinement, presence of elastic multipoles of other orders. The sizes r_a and r_b are relatively inconsequential for a pure multipole behavior (though they can tune its strength). However, our study shows how by varying r_a and r_b we can tune the strength of multipoles of different order within the expansion, which in turn controls the distance and angular dependencies of colloidal interactions – this is explored numerically by means of the multipole expansion (Figs. 5,6), as described in detail in the manuscript and is consistent with the experiments (Fig. 3) for a particular particle selected for demonstration as an example. While we are eager to show more data on various aspects of interactions, we are at maximum allowed number of large figures and have to resort to leaving this for future full-size publications. We appreciate this remark which helped us to carefully review the description of our findings and to assure that the readers can follow it.

4- Regarding Fig. 3h, in line 171, the text reads “...the experimental angular diagram (Fig. 3h) is more complicated, pointing out a richer behavior that can be understood by presence of other non-zero elastic multipoles and consistent with the lack of symmetry plane orthogonal to n_0”. I found it hard to relate the plots provided in Fig. 3h with this text. A detailed explanation of what is reported in the figure and its relation to this text may help the reader.

Authors:

We thank the Referee #1 for bringing this to our attention. We have split the original sentence into several shorter ones and re-wrote these sentences to assure clarity of the statements. We appreciate this remark that directed this improvement.

Referee #1:

5- Figures 5, 6, and 8 are complicated and sometimes hard to follow with the related text. Simplification these figures or placing some pointers regarding the features discussed in the text may help.

Authors:

We thank the Referee #1 for drawing our attention to the difficulty of following some parts of the manuscript. We have introduced features/labels discussed in the text and, as the Referee suggested, pointers that will make it easier for the readers to follow our discussion of these results (e.g. when we discuss conditions under which different multipoles are pronounced as the leading-order or pure multipoles). We appreciate Referee's thoughtful comments that directed these improvements.

Referee #1:

In addition to these, there are several typographical errors that I list below;

Line 184 – "... magnitudes of the ...",

Line 189 – SPMB abbreviation was not defined before,

Line 364 – "... forms an elastic quadrupole..."

Authors:

We thank Referee #1 for noticing these typos and we corrected them in the revised version. This also prompted us to carefully proofread the manuscript and correct several other minor typos, which improves the clarity of the revised manuscript. We are very grateful to the Referee for the very positive report and for the helpful remarks that allowed us to further improve our manuscript.

Report of Reviewer #3:

The work by Smalyukh et.al. studied the creation of elastic multipole by dispersing colloidal dimer particles into small-molecule nematic liquid crystal hosts, with the hypothesis that shapes and surface boundary conditions of the dimers would readily define the types of multipole that can be produced. Through experimental observations as well as numeric simulations, the authors described the realization of various elastic multipoles—in particular the 32- and 64-pole, which was not previously demonstrated. The investigation is systematic, providing the underlying physical principles and an effective means to design colloids with low- and high-order elastic multipoles. Analogies are also made between the multipoles and the orbitals of atoms, which extends the importance of work to both natural and unnatural material structures.

Authors:

We thank the Referee #3 for the positive report, describing our manuscript as “*systematic, providing the underlying physical principles and an effective means to design colloids with low- and high-order elastic multipoles.*” which “*extends the importance of work to both natural and unnatural material structures.*” We also appreciate the remarks below that allow us to further improve clarity of our manuscript and make it even more appropriate for the broad readership of *Nature Communications*.

Reviewer #3:

Although the current study is solid, its significance is of some concern. Previously, the author's group has studied various types of particle shapes in order to obtain the elastic defects and interactions. This includes triangles and squares (science 2009), and knots (Nat. Mat., 2014), which demonstrated lower-order elastic dipoles and quadrupole. Most recently, the hexadecapole was reported in spherical colloids (Nat. Commun. 2016). In some cases, the self-assembly of those colloidal multipoles was demonstrated. While the current study extended the shapes to colloidal dimers with the same or different surface/composition, compared the hexadecapole previously reported, the discovery of 32- and 64-pole seems natural and not as surprising. However, the dimer shapes selected in the current study have their own advantages, which is the continuous tunability for various parameters (diameters, center-center distance, and conic angles, etc.). This has allowed much control over the multipoles one can create since the dominating multipoles are evolving as the shape

changes, as demonstrated by the simulation results. Since colloidal dimers with size/composition/aspect ratio are easily accessible through facile synthesis, the author should consider to conduct a relatively more systematic study to verify the numeric simulation.

Authors:

We thank the Referee #3 for these remarks, which allowed us to better clarify the novelty of our present work. We note that our current study is not just reporting the discovery of 32- and 64-pole, octupole, and other elastic multipoles, but rather the physical principles behind the on-demand realization of elastic multipoles of different, low-to-high order. We then selected a series of examples to demonstrate this to readers, including the 32- and 64-poles (and also octupoles and also now common dipoles, quadrupoles and hexadecapoles) elastic multipoles. To account for this remark, we have clarified throughout the manuscript text the main goal of this work is the demonstration of the physical design principles for obtaining arbitrary-order elastic multipoles, which is tested through comparing experiments and simulations. We agree with the Referee that the use of the colloidal dimer-oligomer system, which can be obtained using wet chemical synthesis in a facile way, will lead to many in-depth explorations of these different elastic multipoles and their self-assembly. We note, however, that the scope of the present study is focused on demonstrating the design principles for obtaining multipoles of desired order, but not the in-depth exploration of any particular type of multipoles (which also would be impossible due to space constrains – we use the maximum allowed number of figures & text for Nat. Commun.).

Reviewer #3:

In addition, the manuscript is somewhat lengthy, and feels like that many stories are presented in a busy manner, ranging from atomic orbitals to metamaterials and telecommunications. While some terms are there without introduction (e.g., $n(r)$, SPMB), and sometimes sentences are repeating (e.g., synthesis of dimers by van Der Waals interaction, and reasons why SPMB is brownish, etc.). Revision should be done.

Authors:

We thank the Referee #3 for this remark, which draws our attention to the need of making the article more accessible to the broad readership. We have now assured that all terms and abbreviations are properly introduced when they appear first time. We have streamlined the article, eliminated redundancy and repetitions, and trust that it is now appropriate for the broad readership. We appreciate Referee's remarks that guided these changes and improvements of our article.

Reviewer #3:

Lastly, colloidal dimers (or Janus particles) are anisotropic particles widely used in the field of colloidal assembly due to directional interactions. The author may consider the demonstration of some simple assembly structures based on the elastic multipole arguments presented, and speculate on how higher order multipole may expand the assemblies of Janus dimer particles.

In summary, although the new shapes studied are desirable in the field, I have some concern about the suitability of it being published in Nat. Commun, given the fact that investigation on multiple other shapes have been reported. Some assembly structures that reflects conclusions of the study should strengthen the paper.

Authors:

We thank the Referee #3 for these suggestions. In the revised version of Figure 3, we have added examples of self-assembled pair assemblies of the colloidal particles we study, which occur when particles approach each other within different sectors of attraction (see Fig. 3 and text for details). We do have significantly more experimental data, but our article is already at the maximum allowed number of figures and words of text per Nature Communications guidelines. Moreover, the Referee himself/herself indicated in the remark above the need of streamlining the manuscript, which we did, but which would be even harder to do when adding more and more data. To summarize, Referee's comments were helpful in better highlighting the novelty of this present work, in particular in emphasizing that the goal of this study is in developing the design principles for obtaining desired elastic multipoles of arbitrary order. We have accounted for all Referee's suggestions, streamlined the article and assured that it is appropriate and accessible for a broad readership, and we trust that it can now be published in its present form.

Report of Reviewer #4:

This is an extremely interesting manuscript that describes experiments and theory that push the paradigm of "colloids as giant atoms" substantially beyond the current status-of-the-art. Of course the field has seen colloidal spheres that mutually interact with spherically symmetric potentials, dipolar interactions by magnetic cores or by high-frequency electric fields, and patchy particles and DNA-coated particles with specific binding sites and binding geometries. However, the present manuscript gives a full detailed recipe to construct multipolar colloid-colloid interactions by coupling (anchoring) the colloidal particles to the nematic director field of a molecular liquid crystal medium. As consequence, monopoles, dipoles, quadrupoles, and much higher multipoles appear, with structures that can be controlled by the anchoring conditions on the colloidal surface and the shape of the colloidal particles. This holds the prospect of a zoo of (self-assembled) structures of which the authors only scratched the surface. The opening of such a field is worthy of being published in Nature Communications. The manuscript is well written and clear, and contains beautiful graphic material. The experiments and the theory are in good harmony, and the good agreement suggests that the system is actually very well understood in terms of nematic elasticity.

Authors:

We thank the Referee #4 for the very positive report, describing our manuscript as “*an extremely interesting manuscript that describes experiments and theory that push the paradigm of "colloids as giant atoms" substantially beyond the current status-of-the-art.*” with “*experiments and the theory are in good harmony.*” We are glad the Referee #4 finds that our manuscript is “*worthy of being published in Nature Communications*” and thank for useful comments. We have accounted for all suggestions of this report (as detailed below) that helped us to significantly improve the overall presentation of our results.

Reviewer #4:

I have a few questions that popped up upon reading the manuscript. Their answer could expand the horizon of this paper even more, if not from an experimental than perhaps from a theoretical perspective:

1. The authors draw analogies with atomic orbitals, which is fascinating, but makes one wonder if there is an elastic analogy with "orbital hybridization"? In atomic systems this phenomenon takes place because of "non-diagonal" matrix elements. Could one for instance envisage a diamond-like structure by s-p³ hybridation as takes place in carbon, with possibly great consequences for photonic bandgap materials? Or likewise for e.g. graphene-like structures?

Authors:

We thank the Referee #4 for this excellent comment! Indeed, the hybridization at the level of atomic orbitals is the result of combination of multiple atomic orbitals (for example for sp³ hybridisation, the s and three p orbitals), and actually similar could be envisaged by creating combination of elastic multipoles. Importantly, the colloidal particles used in this work are rotationally symmetric (along the z axis, as defined in the manuscript) and also impose rotationally symmetric anchoring profile, which results in the fact that all multipole coefficients of these particles are scalars (i.e. single number). However, in full definition, elastic multipoles (coefficients) are tensorial quantities [for example see: V. M. Pergamenschik and V. A. Uzunova, Dipolar colloids in nematostatics: Tensorial structure, symmetry, different types, and their interaction, Phys. Rev. E 83, 021701 (2011)]. Such generalized elastic multipoles can be realized by breaking symmetries of the nematic distortion field around colloidal particles, for example by particle geometry or imposed surface anchoring pattern (i.e. patchy particles), which exactly in clear analogy creates the “non-diagonal” matrix elements, as pointed out by the Referee. Thus, if having controllably realized such tensorial elastic multipoles, one could open routes for hybridization. To include this interesting possibility for hybridisation, text is added on pages 20-21. We also slightly expand the discussion of how different-order elastic multipoles may allow for assembly of colloidal structures that mimic molecules, crystals and even various spin glasses. We note that much of the emphasis in the current work is put on high-order elastic multipoles, which do not have direct analogs in the atomic systems. The Referee is correct that there are interesting analogies in bonding of low-order-multipole nematic colloids and atoms and this, we hope, will be addresses in the future studies. We thank the Referee #4 for this helpful remark.

Reviewer #4:

2. Atomic orbitals are purely electrostatic in nature, stemming from the $1/r$ Coulomb potential in 3D such that the potential satisfies the Laplace equation -not unlike the director field in the present study. However, in colloid science I would always expect impurities and/or surfaces that will give rise to screening; in an electrostatic analogue the length scale is the Debye length, in a superconductor the penetration depth. I guess there must be a liquid analogue as well? I suppose, given the good agreement between theory and experiment in this case, that this "screening length" is much larger than the particle size, but this probably depends on the stiffness of the LC and could perhaps also be tuned experimentally?

Perhaps the authors can comment on these issues in order to put limits on the regime of applicability of their system and ideas. Clearly, however, these answers could also be part of a research agenda that is opened-up by this work, that I recommend for publication.

Authors:

We thank the Referee #4 for the very helpful remark. Indeed, there can be an effect of screening that limits the range of elastic inter-particle interactions caused by confining surfaces of liquid crystal cells. In this case, similar to electrostatics, the interaction with confining surfaces can be modeled using a method of image charges/multipoles, which will be of type and strength determined by the nature of the surface boundary conditions. The characteristic length scale in the case of strong boundary conditions is the sample size. In the case of weak boundary conditions on confining surfaces, the so-called surface anchoring extrapolation length can come into play. We have made brief remarks about this within the revised discussion. We appreciate the Referee's positive report and the recommendation that the article is published.

We believe that the above-described revisions address all of the remarks and account for all suggestions that were raised by the original manuscript. We also feel that they improve the paper significantly. We would like to thank the Referees for the thoughtful valuable comments that helped to direct these changes. We trust that this revised version is now well suited for publication in *Nature Communications* in the present form.

Sincerely,

Prof. Ivan I. Smalyukh, Department of Physics, 390 UCB
Founding Fellow, Renewable and Sustainable Energy Institute (RASEI)
Senior Investigator, Liquid Crystal Materials Research Center (LCMRC)
University of Colorado at Boulder
2000 Colorado Ave., Boulder, CO 80309
Phone: 303-492-7277 (office); Fax: 303-492-2998
Email: Ivan.Smalyukh@colorado.EDU
<http://www.colorado.edu/physics/SmalyukhLab/index.html>
<http://spot.colorado.edu/~smalyukh/>

REVIEWERS' COMMENTS:

Reviewer #1 (Remarks to the Author):

I have read the rebuttal letter that the authors have prepared and checked the manuscript that they have updated. The authors addressed the remarks that I have raised and I think the manuscript now deserves publication in Nature Communications.

Reviewer #3 (Remarks to the Author):

I have read the revised manuscript, and the authors have addressed my concern and clarify the significance and focus of present work. They have also included some of the assembly experiments asked, in Figure 3. In addition, they have fixed typos and the minor issues with terminology introduction, etc.

The authors have incorporated comments and suggestions from other reviewers.

Therefore, I would recommend its publication on Nature communication.

Reviewer #4 (Remarks to the Author):

The authors have taken essentially all my comments and suggestions into account, and also added quite a few other improvements to the manuscripts that were suggested by the other referees. On the basis of my previous report and the revisions, I recommend publication of the revised manuscript.

Response to Reviewers comments

Reviewer #1 (Remarks to the Author):

I have read the rebuttal letter that the authors have prepared and checked the manuscript that they have updated. The authors addressed the remarks that I have raised and I think the manuscript now deserves publication in Nature Communications.

Reviewer #3 (Remarks to the Author):

I have read the revised manuscript, and the authors have addressed my concern and clarify the significance and focus of present work. They have also included some of the assembly experiments asked, in Figure 3. In addition, they have fixed typos and the minor issues with terminology introduction, etc. The authors have incorporated comments and suggestions from other reviewers. Therefore, I would recommend its publication on Nature communication.

Reviewer #4 (Remarks to the Author):

The authors have taken essentially all my comments and suggestions into account, and also added quite a few other improvements to the manuscripts that were suggested by the other referees. On the basis of my previous report and the revisions, I recommend publication of the revised manuscript.

Authors

We are pleased that all Referees recommended our revised manuscript for publication in *Nature Communications*.